**Spatial and temporal variability of sea-salts in ice/firn**
**cores from Fimbul Ice Shelf, Dronning Maud Land – DML,**
**Antarctica**
Carmen Paulina Vega,[1,2,¶,§] Elisabeth Isaksson,[1] Elisabeth Schlosser,[3,4] Dmitry
Divine,[1] Tõnu Martma,[5] Robert Mulvaney,[6] Anja Eichler,[7] and Margit Schwikowski-
Gigar.[7]
[1]{Norwegian Polar Institute, N-9296 Tromsø, Norway}
[2]{Department of Earth Sciences, Uppsala University, Villavägen 16, SE-752 36,
Uppsala, Sweden}
[3]{Institute of Atmospheric and Cryospheric Sciences, University of Innsbruck,
Innsbruck, Austria}
[4]{Austrian Polar Research Institute, Vienna, Austria}
[5]{Department of Geology, Tallinn University of Technology, Tallinn, Estonia}
[6]{British Antarctic Survey, Madingley Road, High Cross, Cambridge,
CambridgeshireCB3 0ET, United Kingdom}
[7]{Paul Scherrer Institute, 5232 Villigen PSI, Switzerland}
Now at:
[¶] {School of Physics, University of Costa Rica, San Pedro de Montes de Oca,
11501-2060 San Jose, Costa Rica}
[§] {Centre for Geophysical Research, University of Costa Rica, San Pedro de
Montes de Oca, 11501-2060 San Jose, Costa Rica}
Correspondence to: C. P. Vega (carmen.vegariquelme@ucr.ac.cr)
**Abstract**
Major ions were analysed in firn/ice cores located at Fimbul Ice Shelf (FIS), Dronning
Maud Land – DML, Antarctica. FIS is the largest ice shelf in the Haakon VII Sea,
with an extent of approximately 36 500 km$^2$. Three shallow firn cores (about 20 m
deep) were retrieved in different ice-rises, Kupol Ciolkovskogo (KC), Kupol
Moskovskij (KM), and Blåskimen Island (BI), while a 100 m long core (S100) was

drilled near the FIS edge. These sites are distributed over the entire FIS area so that they provide a variety of elevation (50–400 m a.s.l.) and distance (3–42 km) to the sea. Sea-salt species (mainly $Na^+$ and $Cl^-$) generally dominate the precipitation chemistry in the study region. We associate a significant six-fold increase in sea-salts, observed in the S100 core after the 1950s, with enhanced sea-salt aerosol production from blowing salty snow over sea-ice. This increase in sea-salt concentrations is synchronous with a shift in non-sea-salt sulfate ($nssSO_4^{2-}$) toward negative values, suggesting a possible contribution of fractionated aerosol to the sea-salt load in the S100 core most likely originating from salty snow found on sea-ice. In contrast, there is no evidence of a significant contribution of fractionated sea-salt to the ice-rises sites, where the signal would be most likely masked by the large inputs of biogenic sulfate estimated for these sites. In summary, these results suggest that the S100 core contains a sea-salt record dominated by processes of sea-ice formation in the neighbouring waters. In contrast, the ice-rises firn cores register the larger-scale signal of atmospheric flow conditions and a less efficient transport of sea-salt aerosols to these sites. These findings are a contribution to the understanding of the mechanisms behind sea-salt aerosol production, transport and deposition at coastal Antarctic sites, and for the improvement of the current Antarctic sea-ice reconstructions based on sea-salt chemical proxies obtained from ice cores.

## 1 Introduction

Antarctic ice and firn cores contain valuable information about the climate and atmospheric chemical composition of the past and provide evidence for the important role of Antarctica in the global climate system. Numerous ice and firn cores have been drilled in Antarctica during the past decades (Stenni et al., 2017). However, relatively few cores were drilled in coastal regions, which are more sensitive to changes in climate than the dry and cold interior of Antarctica. In fact, two recent review papers point out the lack of ice core data from low elevation coastal areas when discussing Antarctic climate variability (Stenni et al., 2017; Thomas et al., 2017). In an effort to understand the role of ice shelves in stabilizing the Antarctic ice sheet, particular focus has been laid on the investigation of ice-rises and ice rumples as buttressing elements within the ice sheet - ice shelf

complex (Paterson, 1994; Matsuoka et al., 2015). Furthermore, due to their radial ice flow regime, generally low ice velocities, and relatively high surface mass balance (SMB), ice-rises are potentially useful sites for ice core retrieval (Philippe et al., 2016; Vega et al., 2016). Firn and ice cores drilled at ice-rises allow obtaining high-resolution climate records to investigate sub-annual and long-term temporal changes in the loads of different chemical compounds found in the snow, providing information about their sources and transport, particularly of sea-salt ions, such as sodium ($Na^+$) and chloride ($Cl^-$), which are strongly modulated by sea-ice extent and meteorological conditions. Recent modelling efforts to study the use of sea-salts as proxies for past sea-ice extent have shown that, under present climate conditions and on interannual timescales, meteorological conditions rather than sea-ice extent are the dominant factor modulating atmospheric sea-salt concentrations that are deposited at the interior and coastal sites in Antarctica (Levine et al., 2014). However, sea-salts have the potential as proxy for sea-ice extent at glacial-interglacial scales when large changes in sea-ice extent took place (Levine et al., 2014).

At most Antarctic sites, atmospheric sea-salt concentrations present maxima during austral winter (Wagenbach et al., 1998; Weller and Wagenbach, 2007; Jourdain et al., 2008; Udisti et al., 2012), with the exception of Dumont D'Urville where maxima occur during summer (Wagenbach et al., 1998). Similarly, sea-salt fluxes obtained from Antarctic ice cores also show winter maxima (Abram et al. 2013 and references therein). However, in some recent core records from coastal sites no clear seasonality is observed, e.g. at Mill Island during the period 1934–2000 (Inoue et al., 2017). Abram et al. (2013) conclude that despite the seasonal signal registered in different Antarctic ice cores, sea-salt fluxes do not show a consistent relationship with sea-ice extent on inter-annual timescales, and on the contrary, are highly dependent on atmospheric transport, and/or the presence of polynyas.

Hitherto, two main sources of increased winter sea-salt aerosols have been proposed: (i) increased storminess leading to an enhancement of sea-salt aerosols above the open ocean with possibly faster meridional transport (Petit et al., 1999; Fischer et al. 2007), and (ii) a direct input of sea-salts associated to increases in

sea-ice, overcoming source (i), e.g. due to frost flowers (Rankin and Wolff, 2002; Rankin et al., 2004; Roscoe et al., 2011), brine (Rankin et al., 2000), and the contribution of snow transported over sea-ice by wind (Yang et al., 2008, 2010; Huang and Jaeglé, 2017; Rhodes et al., 2017).

In the review by Abram et al. (2013), the authors suggest that the brine-frost flower system is a plausible source of sea-salt aerosols to coastal Antarctic sites. This hypothesis is supported by the experimental evidence that the original seawater $SO_4^{2-}/Na^+$ ratio cannot be used in the non sea-salt sulfate ($nssSO_4^{2-}$) calculations, leading to negative $nssSO_4^{2-}$ values both in winter aerosol and fresh snow sampled at coastal sites (Hall and Wolff, 1998; Wagenbach et al., 1998; Curran et al., 1998; Rankin and Wolff, 2002 and 2003), and also in ice cores from both inland (Wagenbach et al., 1994, Kreutz et al., 1998) and coastal sites (Inoue et al., 2017). These negative values indicate that a lower $SO_4^{2-}/Na^+$ ratio has to be used in $nssSO_4^{2-}$ calculations, i.e., a depletion of $SO_4^{2-}$ with respect to seawater composition, occurred in wet and dry deposition.

During the process of sea-ice formation, ions present in the water are not incorporated in the ice crystal matrix, but remain as highly concentrated brine in brine pockets or channels. The brine can be transported by capillary effects through brine channels to the newly formed ice surface, resulting in a thin layer of highly saline surface brine. This fractionated brine is unlikely to be a direct source of sea-salts because it usually quickly gets covered by snow, and no clear mechanism has been found to explain how this brine could become airborne (Abram et al., 2013). With further cooling of the ice, the volume of brine decreases and consequently, its salinity increases, leading to the precipitation of different saline compounds. This depends on temperature, e.g. sodium sulfate or mirabilite ($Na_2SO_4 \cdot 10\,H_2O$) starts to precipitate at temperatures below −8 °C, and sodium chloride (NaCl) at temperatures below −26 °C. Consequently, the remaining brine is depleted in sodium and sulfate ions via precipitation of mirabilite at relatively mild polar temperatures. Frost flowers can form from this brine when meteorological conditions are adequate, i.e. at low intensity winds, which allows these delicate structures to grow without breaking apart, and on very thin ice where a strong temperature

gradient is present between the ice surface and the overlying air (Rankin et al., 2000; Rankin and Wolff, 2002, and references therein). Thus, frost flowers formed at temperatures below −8 °C will be depleted in sodium and sulfate relative to other ions present in seawater (Rankin et al., 2000; Rankin and Wolff, 2002), evidenced by negative $nssSO_4^{2-}$ values measured in aerosols and snow (see section 2.3 for more details on the calculation of the nss-fractions).

For most of the last decade, frost flower formation, transport and deposition, has been considered the most plausible mechanism behind the fractionated aerosol detected at coastal areas. However, Yang et al. (2008 and 2010), and Huang and Jaeglé (2017) proposed an alternative mechanism: the origin of sea-salt aerosol could be due to the sublimation of blowing salty snow. This salty snow could be a result of frost flower formation, upward migration of brine within the snow (Massom et al., 2001), or by the input of sea-spray from the open ocean or nearby leads or polynyas (Dominé et al., 2004). Flooding of sea-ice under the weight of accumulated snow can also induce increased salinity of snow (Massom et al., 2001). As the snow can be contaminated or wetted with fractionated brine or frost flowers, it could be expected that this salty snow also shows such fractionation. As pointed by Yang et al. (2008 and 2010), this salty snow can be transported by wind and if the air is not saturated, the snow particles may lose water by sublimation and become sea-salt aerosols. These aerosols could then be transported and deposited either by dry or wet deposition, depending on local meteorology.

According to Abram et al. (2013), the idea proposed by Yang et al. (2008) is plausible for coastal sites, along with the frost flower mechanism. Consequently, snow present on new sea-ice and frost flowers are important features that, combined with wind transport, need to be taken into account when interpreting the sea-salt record of coastal ice and firn cores.

This study discusses sub-annual and long-term temporal changes in sea-salt and major ion concentration measured in three recently drilled firn cores from different ice-rises located at Fimbul Ice Shelf (FIS): Kupol Ciolkovskogo, Kupol Moskovskij, and Blåskimen Island, a 100 m long core drilled near the FIS edge (S100), and five snow pits (Table S1, Supplementary material) sampled along the ice shelf

(Figure 1). The main goals of the present study are to investigate possible mechanisms behind deposition, sub-annual, and spatial variability of sea-salts in this coastal region. The results presented here contribute to bridging the data gap existent at coastal Antarctic sites, and to the improvement of current Antarctic sea-ice reconstructions based on sea-salt chemical proxies.

## 2   Methods

### 2.1   Study area

With an extent of approximately 36 500 km$^2$, FIS is the largest ice shelf in the Haakon VII Sea (Figure 1). Fed by Jutulstraumen, the largest outlet glacier in DML, FIS is divided into a fast moving ice tongue, Trolltunga, directly feeding the central part of the ice stream, and slower surrounding parts. Several ice-rises (250–400 m a.s.l.; 10–42 km from the coast) are found at FIS, varying in size from 15 to 1200 km$^2$, and located approximately 200 km apart.

Early investigations in this area began during the International Geophysical Year (IGY) 1956/57 (Swithinbank, 1957; Lunde, 1961; Neethling, 1970) and continued during the last decades with focus on surface mass balance (SMB) variability in space and time (Melvold et al., 1998; Melvold, 1999; Rolstad et al., 2000; Isaksson and Melvold, 2002; Kaczmarska et al., 2004; Kaczmarska et al., 2006; Divine et al., 2009; Sinisalo et al., 2013; Schlosser et al., 2012, 2014; Langley et al., 2014; Vega et al., 2016). However, studies on spatial and temporal variability of chemical composition of snow and ice from this area are limited to water stable isotopes interpretations (Kaczmarska et al., 2004; Schlosser et al., 2012, 2014; Vega et al., 2016).

SMB obtained from the S100 core (Figure 1) retrieved at FIS shows a mean long-term accumulation rate of 0.3 m water equivalent per year (m w.e. yr$^{-1}$) for the period 1737–2000, with a significant negative trend in SMB for the period 1920–2000 (Kaczmarska et al., 2004). This negative trend in SMB has been reported in several shorter firn cores from the region (Isaksson and Melvold, 2002; Divine et al., 2009; Schlosser et al, 2014), including one record from the Kupol Ciolkovskogo ice-rise (Vega et al., 2016).

More detailed information on previous campaigns, glaciological and meteorological conditions at FIS and the core sites at the ice-rises, can be found in Vega et al. (2016) and Goel et al. (2017), and references therein, whereas an overview on Antarctic ice-rises is given in Matsuoka et al. (2015).

## 2.2 Sampling

Three shallow firn cores (about 20 m deep) were retrieved at different ice-rises (Kupol Ciolkovskogo (KC), Kupol Moskovskij (KM), and Blåskimen Island (BI), Figure 1, Table 1), located at FIS between January 2012 and January 2014 during field expeditions organized by the Norwegian Polar Institute (NPI). Location, elevation, and length of the different ice-rises cores are presented in Table 1. Each core was drilled from the bottom of a 2 m snow pit (not sampled for major ions). The firn density was determined as bulk density of each sub-core piece (average length of 45 cm), and of each snow pit interval (20 cm). The samples were collected following clean protocols (Twickler and Whitlow, 1997), shipped frozen to NPI, and later to the Paul Scherrer Institute (PSI), Switzerland, for cutting and chemical analysis. Sample resolution varied between 4 and 8 cm depending on sample depth and density. Thickness of ice lenses, water stable isotope ratios and SMB for the three ice-rises are reported in Vega et al. (2016). Additionally, unpublished major ion concentrations measured in the 100 m deep S100 core drilled in austral summer 2000/2001 (Kaczmarska et al., 2004) were included in this study (Figure 1, Table 1). The S100 core was sampled at 5 cm resolution between top and 6 m deep, and then at 25 cm resolution between 6 m to 100 m deep.

## 2.3 Chemical analyses

Major ions (methanesulfonic acid (MSA), $Cl^-$, $NO_3^-$, $SO_4^{2-}$, $Na^+$, $K^+$, $Mg^{2+}$ and $Ca^{2+}$) present in the three firn cores from the ice-rises were analysed at PSI using a Metrohm ProfIC 850 ion chromatograph combined with an 872 Extension Module and auto-sampler. The precision of the method was within 5 % and detection limits (D.L.) were below 0.02 $\mu$mol $L^{-1}$ for each ion (Wendl et al., 2014). Ion concentrations (MSA, $Cl^-$, $NO_3^-$, $SO_4^{2-}$, $Na^+$, $K^+$, $Mg^{2+}$ and $Ca^{2+}$) in the S100 core were measured

at the British Antarctic Survey (BAS) using fast ion chromatography (Littot et al.,
2002). The reproducibility of the measurements was 4–10 %.
$Cl^-$, $SO_4^{2-}$, $K^+$, and $Mg^{2+}$ non sea-salt fractions (nss) were calculated from the mean
seawater composition using the sea-salt $Na^+$ fraction ($ssNa^+$) as standard ion
(section 3.5), using:
$$[\text{nssX}]=[\text{X}]_{\text{total}} - k \times [\text{ssNa}^+] \ \ ,$$
where
$$k = \frac{[\text{X}]_{\text{seawater}}}{[\text{Na}^+]_{\text{seawater}}},$$
using the standard mean chemical composition of seawater with ion concentration
expressed in $\mu$mol $L^{-1}$ ($k$ values are listed in Table S2), and where
$$[ssX] = k \times [ssNa^+]$$
Due to the low concentrations of $NO_3^-$ in standard seawater (Summerhayes and
Thorpe, 1996), $NO_3^-$ was not separated into nss- and ss-fractions (i.e., $NO_3^-$ was
assumed to have a nss-origin only, as well as MSA). The $nssNa^+$ and $ssNa^+$
fractions were calculated using $Ca^{2+}$ as reference ion and $k$= 1.40 for Earth's crust
composition (Lutgens and Tarbuck, 2000) (section 3.5).
In addition, water stable isotopes analyses of the KC, KM and BI cores are described
in Vega et al. (2016); while analysis of the S100 core is described in (Kaczmarska
et al., 2004).

## 20   **2.4   Firn and ice core timescales**

The timescales of the KM and BI cores were obtained based on annual layer
counting of water stable isotope ratios ($\delta^{18}O$), and found to cover the periods
between austral winter-1995(96) and summer-2014, respectively. The error in the
dating was estimated as ± 1 year for both of these cores (Vega et al., 2016). Both
KC and the S100 cores were dated using a combination of annual layer counting of
$\delta^{18}O$ and identification of volcanic horizons (i.e. by using the $SO_4^{2-}$, dielectric
profiling (DEP), and electrical conductivity measurements (ECM)), with timescales
covering the time period 1958–2012 (± 3 years) at KC (Vega et al., 2016), and 1737–
2000 (± 3 years) at S100 (Kaczmarska et al., 2004).

# 3 Results

## 3.1 Ion concentrations and sources

Median, mean, maximum, minimum, and standard deviation ($\sigma$) of concentration for all ions measured in the cores are shown in Table 2. Box-plots of raw ion concentrations in the different cores are shown in Figure S1 in the Supplementary material. In addition, median, mean, maximum, minimum, and $\sigma$ of concentrations for all ions measured in the FIS snow pits are shown in Table S3, while boxplots are shown in Figures S2 and S3 in the Supplementary material.

In general, concentrations in the KM core are higher than in the other ice-rises cores, and snow pits, e.g. about eight-fold higher concentrations of $Na^+$, $K^+$, $Mg^{2+}$ and $Cl^-$ in KM than in the KC core are found for the period 1995–2012. The relatively high $Na^+$ and $Cl^-$ concentrations observed in the KM core are also detected in the upper meters of the S100 core (in the periods 1995–2000, and 1950–2000, respectively, Table 2). Similarly high values have been reported in several snow and firn samples from other western DML coastal sites (Kärkäs et al., 2005), and in Mill Island, Wilkes Land (Inoue et al., 2017). Ion concentrations in the snow pits (Table S3) are in reasonable agreement with firn and ice core values and with ion concentration ranges for snow pits previously sampled at FIS (Mulvaney et al., 1993). Temporal and spatial variability of ion concentrations are explored in more detail in the following sections.

In order to assess the most important sources explaining the total variance in the glacio-chemical records from FIS, a principal component analysis (PCA) was applied to the different ion series measured at the KC, KM, BI, and S100 cores. Years, in which no sub-annual concentrations were available in the S100 (1793, 1841, 1866, 1918, and 1944) due to low resolution, were filled in by linearly interpolating between the annual means of the previous and following year. For the PCA analysis, the logarithms of the raw concentrations were used (at sub-annual (using the raw values as input) and annual resolutions) and standardized by subtracting the mean of the data series from each data point and then dividing the result by the standard deviation of the data series. Due to the sampling resolution, only the KM and BI cores were comparable at a sub-annual level. PCA analyses

were performed for three different periods of the S100 core: for the entire time
interval spanning 1737–2000, for the subsection between 1737–1949, and between
1950–2000.
The sum of the variances of the first three principal components (PC1, PC2 and
PC3) was ≥ 80 % of the total variance of the original sub-annual and annual data in
all cores. Since the results of the sub-annual and annual PCA analysis are similar
only the annual results are considered. The loadings of the first three (KC) and two
(KM, BI, and S100) principal components are shown in Table 3. PCA results are
consistent between the different cores. Consequently, the ions can be separated in
two main groups: sea-salts species ($Na^+$, $Cl^-$, $K^+$, $Mg^{2+}$, and $Ca^{2+}$) and marine-
biogenic/mixed (MSA, $SO_4^{2-}$, including $NO_3^-$) (Table 3).
Generally, our results indicate that the major sources of the ions at the different sites
are the same, independent of the core site and mean concentrations of ions in the
cores. Only at the KC site the PCA results imply an additional input of $Ca^{2+}$ from
other sources than sea-salt, as for instance mineral dust. High loadings of $NO_3^-$ and
MSA in PC2, and thus, coherence between both species, have been observed in an
ice core from Lomonosovfonna, Svalbard (Wendl et al., 2015), and a fertilizing effect
was proposed as explanation for those findings. Wendl et al. (2015) suggest that
enhanced atmospheric $NO_3^-$ concentrations and the corresponding nitrogen input
to the ocean can trigger the growth of dimethyl-sulfide-(DMS)-producing
phytoplankton. However, there is a variety of possible $NO_3^-$ sources to polar sites,
and the relative importance of these sources at certain locations and time is still in
discussion (Mulvaney and Wolff, 1993; Savarino et al., 2007; Wolff et al. 2008;
Weller et al. 2011; Pasteris et al., 2014; Sofen et al. 2014).
**3.2   Long-term variability of ion concentrations**
We use the two longest available records for FIS (KC and S100) to explore the long-
term temporal variability of major ions, with special focus on sea-salts, represented
by $Cl^-$ and $Na^+$ (Figure 2). In the S100 core, $Na^+$, $Cl^-$, $K^+$, and $Mg^{2+}$ median
concentrations show a marked six-fold increase after the 1950s. However, there is
no significant increase of the concentration of these species in the KC core. Due to
its limited time coverage it cannot be determined if there was a substantial relative

increase in concentrations at this site after the 1950s. MSA and $NO_3^-$ concentrations do not show such marked increase in the S100 core and values agree between both cores after the 1950s (Figure S4 in the Supplementary material). Consequently, three periods can be distinguished in the S100 record: (i) the period between 1995–2000, comparable to the time covered by the KM and BI cores; (ii) the period between 1737–1949, where ion concentrations remain low; and, (iii) the period between 1950–2000, where sea-salt concentrations increased (Table 2).

With the exception of MSA, all ions show a positive trend (significant at the 95 % confidence level) during the period 1950–2000, although the slope for $NO_3^-$ is three orders of magnitude smaller than for the other ions (slope and error of the linear regression are shown in Table S4 in the Supplementary material). Such significant linear trend was not observed in the KC ion record over the same period.

Ions, with the exception of MSA, also show a positive and significant trend between 1737–1949, (Table S4), however, the increase is less marked than during the 1950–2000 period.

## 3.3  Sub-annual variability of ion concentrations

The lack of extensive precipitation measurements at sub-annual resolution near the sampling sites at FIS, makes a precise reconstruction of the precipitation regime at the area difficult. To obtain a time scale for the KC, KM, and BI ice-rises cores, Vega et al. (2016) employed $\delta^{18}O$ winter minima and summer maxima, and assumed uniform precipitation throughout the year at the core sites. The assumption was made on the basis of precipitation data for DML reported by Schlosser et al. (2008), which showed high temporal variability in the monthly sums due to the influence of cyclone activity affecting both, coastal and inland regions. In addition, at Neumayer station (70º 39' S, 8º 15' W), the closest to the ice-rises core sites, two precipitation maxima (April and October) are identifiable for the period 2001–2006, possibly a manifestation of the semi-annual oscillation of the circumpolar trough (Schlosser el al., 2008). Considering the above, to investigate the sub-annual variability of the different ion groups in the KM, BI, and S100 cores, we associated the winter minima and summer maxima in $\delta^{18}O$ determined in the KC, KM, and BI cores (Vega et al.,

2016), and in the S100 core (Kaczmarska et al. ,2004), with the months of July and January, respectively. The values for April and October were derived by interpolation between January–July, and July–January, respectively, in each core time scale. We defined *summer samples*, as samples within November and April (NDJFMA), and *winter samples*, as samples within May and October (MJJASO). Summer and winter mean concentrations were then calculated based on logarithms of raw ion concentrations expressed in $\mu mol\ L^{-1}$. Ion concentrations were not available at the top 2 m (removed before drilling), therefore, the composite year consisted of 16 (1996–2011) and 15 (1997–2011) complete years for the KM and BI cores, respectively. In the S100 core, sub-annual variability were investigated only during the period 1995–2000, where the concentrations have sufficient temporal resolution. The resulting summer and winter mean concentrations in the cores are presented in Figure 3.

Sea-salt species ($Na^+$ and $Cl^-$, Figure 3a) show lower concentrations during summer in the BI, and S100 core, whereas in the KM core summer and winter show similar means. Both $Mg^{2+}$ and $Ca^{2+}$ (Figure 3b) show similar means in both summer and winter. MSA concentrations (Figure 3c) show summer maxima in all three cores, with a higher summer to winter difference in the BI core, compared with the KM, and S100 cores. These summer maxima are in agreement with the main source of MSA (marine-biogenic), most active during the warmer months. The MSA winter minimum is not as pronounced in the KM core as in the BI core, while the lowest MSA minimum is reached in the S100 core. $NO_3^-$ and $SO_4^{2-}$ concentrations (Figure 3d) show a distinct increase toward the summer in the BI core, which is also observed in the KM core, although less defined. KM, and BI $SO_4^{2-}$ concentrations are higher in the summer, while both $NO_3^-$ and $SO_4^{2-}$ summer and winter means are similar in the S100 core.

## 3.4  Ions spatial variability

In order to investigate ion spatial variability at FIS, we used median annual ion concentrations in the different ice-rises cores (KC, KM and BI), and S100 for the overlapping period between 1997 and 2000, and compared them with latitude,

longitude, site elevation, and distance from the sea (obtained from the GIS package Quantarctica, www.quantarctica.org) (Table 4).

Only annual $SO_4^{2-}$ and annual MSA concentrations show a significant decrease (at the 95 % confidence level) with latitude, and east longitude, respectively. No significant relationship is found between the median annual ion concentrations and latitude, site elevation, and distance from the sea for any of the species. These findings contrast with previous studies from western Dronning Maud Land (WDML) (Stenberg et al., 1998), where a strong correlation between sea-salt concentrations and distance from the sea was found in this area. We attribute the lack of significance for the correlations presented in Table 4 to the local effects on annual SMB due to topography and local meteorology at the KM and BI sites, reported by Vega et al. (2016).

## 3.5 Sea-salt and non sea-salt fractions

PCA results presented in section 3.1 show two main groups in which ions can be separated: sea-salts (ss-fraction), and marine biogenic/mixed (nss-fraction). In order to confirm the common sea-salt source for $Na^+$ and $Cl^-$, we calculated the $Cl^-/Na^+$ ratio, and ion sea-salt and non sea-salt fractions. Table 5 shows the $Cl^-/Na^+$ ratio (expressed in $\mu mol\ L^{-1}$) in the KC, KM, BI, and S100 cores. Medians in the ice-rises cores are equal (KC, and BI) or slightly higher (KM) than the expected ratio in sea water (i.e., $Cl^-/Na^+$= 1.2), while $Cl^-/Na^+$ medians in the S100 core are lower than the expected ratio in sea water, both before and after 1950. Maxima in the ratio vary between 1.5–3.8, and minima between 0.1–0.9. These results show a clear difference in the $Cl^-/Na^+$ ratios between the ice-rises cores and the S100 core, i.e. a $Cl^-$ to $Na^+$ unbalance in the S100 core associated to an excess of $Na^+$. This excess of $Na^+$ can be due to the recombination of biogenic $SO_4^{2-}$ with $ssNa^+$, and/or to additional $nssNa^+$ sources (Legrand and Delmas, 1988). This unbalance can be enhanced by a depletion of $Cl^-$ due to shorter sea-salt atmospheric residence times, and HCl loss from snow (Legrand and Delmas, 1988; Wagnon et al., 1999). HCl loss becomes significant at relatively low snow accumulation rates (Röthlisberger et al., 2003; Benassai et al., 2005), below the accumulation rate reported for the S100

site, therefore, it is unlikely that HCl loss is a dominant factor that could account for the low $Cl^-/Na^+$ ratios at this site. $Cl^-$ depletion by recombination of $ssCl^-$ with atmospheric acids is dependent on the acidic condition of the atmosphere, especially sulfuric acid ($H_2SO_4$), linked to marine biogenic emissions. Due to the seasonality of sulphur biogenic emissions in polar regions, it is expected that the $Cl^-/Na^+$ ratio would present lower values predominantly during the summer months compared to the winter season (Jourdain and Legrand, 2002). Sub-annual $Cl^-/Na^+$ ratios (estimated as explained in section 3.3) in the S100 core show values of $1.4 \pm 0.5$ for the winter period, and $1.2 \pm 0.1$ for the summer period. Since the temporal resolution of the S100 core only allows sub-annual values for the period 1995–2000, is not possible to assess a sub-annual patter on the $Cl^-/Na^+$ ratio, and $Cl^-$ depletion by acidification cannot be ruled out as mechanism to explain the low ratios registered in the S100 core during the last centuries. In addition to $Cl^-$ loss, low $Cl^-/Na^+$ ratios can also be a product of excess $Na^+$ from non sea-salt sources ($nssNa^+$), as for example crustal material from snow-free coastal areas, nunataks, or dust transported from other continents. In order to estimate $nssNa^+$, $ssNa^+$, and the percentage of crustal $nssNa^+$ to total $Na^+$, we used $Ca^{2+}$ as reference ion, therefore, assuming $Ca^{2+}$ only has a crustal origin (Mahalinganathan et al., 2012) and using a $Na^+/Ca^{2+}$ ratio of 1.40 (with concentrations expressed in $\mu mol\ L^{-1}$) for Earth's crust (Lutgens and Tarbuck, 2000). This assumption will introduce an overestimation of the $nssNa^+$ fraction proportional to the ratio $Ca^{2+}/Na^+ = 0.02$ (with concentrations expressed in $\mu mol\ L^{-1}$) in standard seawater, that is not considered when $Ca^{2+}$ is assumed to only have crustal origin. This procedure offers an alternative to obtain $nssNa^+$, without using $Cl^-$ as reference ion, and the ratio $Na^+/Cl^-$ in bulk seawater. Table 5 shows the $nssNa^+$, $ssNa^+$, and percentage of $nssNa^+$ to total $Na^+$ in the different cores. Since some of the calculated $ssNa^+$ values in the KC core were negative (5 %), $ssNa^+$ statistics in Table 5 are shown considering all data points, and only positive $ssNa^+$ values. The KC core presents the largest contribution of $nssNa^+$ to total $Na^+$ with a 21 % in comparison to the KM, BI, and S100 cores (3 %, 4 %, and 5 %, respectively), which is in agreement with PC3 in Table 3 pointing to a strong source of $Ca^{2+}$ to the KC site.

As mentioned in section 2.3, we used the ssNa$^+$ fraction obtained above to calculate nss- and ss-fractions for Cl$^-$, SO$_4^{2-}$, K$^+$, and Mg$^{2+}$ (Table 6). The sea-salt fraction clearly dominates in all ions, with the exception of SO$_4^{2-}$ in the KC, and BI cores, which shows almost three times more nssSO$_4^{2-}$ than ssSO$_4^-$. Nss-fractions often have negative values which can be associated to an ssNa$^+$ enrichment or depletion of major ions in comparison to bulk seawater, i.e. ion fractionation. Negative nss-fractions represent a higher percentage of total values at the S100 core compared to the ice-rises cores, with values up to 90 % for the S100 (1950–2000), and up to 43 % for the KM core.

### 3.6  Evidence for increased fractionated nss-SO$_4^{2-}$ after 1950s

The nssSO$_4^{2-}$ fraction contains all SO$_4^{2-}$ sources besides sea-salts, e.g. marine biogenic emissions, and volcanic emissions. In coastal regions, most of the nssSO$_4^{2-}$ can be attributed to marine biogenic activity via DMS oxidation (Legrand et al., 1992) with maxima in concentrations during the summer (Minikin et al., 1998). To evaluate if ion fractionation is evidenced in the core records, i.e. nssSO$_4^{2-}$ is strongly depleted in SO$_4^{2-}$ relative to Na$^+$ (Rankin and Wolff, 2002), it is necessary to account for the biogenic contribution to total nssSO$_4^{2-}$ at each core. Legrand and Pasteur (1998) have estimated MSA/nssSO$_4^{2-}$ ratios of 0.18 (annual), 0.29 (summer), and 0.86 (winter) (with concentration in $\mu$mol L$^{-1}$) in aerosol collected at Neumayer station, Antarctica. Median MSA/nssSO$_4^{2-}$ ratios calculated in the KC, KM, BI, and S100 cores (Table 7) span a range between 0.1 and 0.3, therefore, closer to the annual and summer values reported by Legrand and Pasteur (1998). Using an annual MSA/nssSO$_4^{2-}$ ratio of 0.18 (Legrand and Pasteur, 1998) and the MSA concentrations measured in the KC, KM, BI, and S100 cores, we estimated the biogenic portion of nssSO$_4^{2-}$ (bio-nssSO$_4^{2-}$) in all the cores to assess the percentage of bio-nssSO$_4^{2-}$ to total SO$_4^{2-}$ (Table 7). In the KM and BI cores, the estimation of bio-nssSO$_4^{2-}$ surpasses the total SO$_4^{2-}$ observed in these cores, while in the KC core the bio-nssSO$_4^{2-}$ would represent about 50 % of total SO$_4^{2-}$. These high percentages were expected especially in the KC, and BI cores, in which the nssSO$_4^{2-}$ fraction dominates over ssSO$_4^{2-}$ (section 3.5). In the S100 core, bio-

nssSO$_4^{2-}$ varies according to the time period considered with percentages three
times higher during the period 1737–1749 (72 %), than the period 1950–2000
(24 %). It is important to bear in mind the estimation of bio-nssSO$_4^{2-}$ when assessing
the possible effect of fractionated aerosols as a source of sea-salts to the snow. In
the ice-rises cores, the high estimated bio-nssSO$_4^{2-}$ percentages would most likely
mask any ssSO$_4^{2-}$ depletion in sea-salt aerosols, making fractionation hard to
evidence; consequently, fewer negative nssSO$_4^{2-}$ values or the absence of them in
the ice-rises cores would not directly indicate that there is no SO$_4^{2-}$ fractionation in
sea-salt found in snow but rather reflect the dominance of bio-nssSO$_4^{2-}$ in these
sites. In the S100 core, this could be relevant for the pre-1950 period in which
estimated bio-nssSO$_4^{2-}$ accounts for 72 % of total SO$_4^{2-}$.
In order to evaluate the possible effect of fractionated aerosols as a source of sea-
salts to the snow on FIS, we used the nssSO$_4^{2-}$ fraction calculated as described in
section 2.3, using $k$ values of 0.06 (Table S2, Supplementary material). The
percentage of nssSO$_4^{2-}$ relative to total SO$_4^{2-}$ is one and a half- to three-times higher
in the KC core than in the other ice-rises cores, KM and BI. Negative median
nssSO$_4^{2-}$ values were obtained in the S100 core, and snow pits M1, M2, and G3
(not shown), with negative nssSO$_4^{2-}$ values being more pronounced after the 1950s.
These negative values found in the snow, i.e. the sea-salt content in snow is strongly
depleted in ssSO$_4^{2-}$ relative to seawater composition, suggest a possible role of
frost flowers and wind-blown salty snow as source of sea-salts (Rankin and Wolff,
2002) to the S100 core (Figure 4a y b, black line). To assess the degree of
fractionation of ssSO$_4^{2-}$ in the cores in respect to seawater, we obtained the linear
regression between annual nssSO$_4^{2-}$ (both positive and negative nssSO$_4^{2-}$ data
points) and annual ssNa$^+$ for the periods 1737–2000 (Figure 5), 1737–1949, and
1950–2000, using a robust regression method that is known to be less sensitive to
a possible heteroscedasticity and non-Gaussianity of the model residuals (which is
a common problem for ion concentration data) than the usual least squares method.
We obtained negative slope values of 0.04, 0.03, and 0.04 for the 1737–2000,
1737–1949, and 1950–2000 periods, respectively. Figure 5 shows a scatter plot of
annual nssSO$_4^{2-}$ vs. ssNa$^+$ for the 1737–2000 period. Following the approach by
Wagenbach et al. (1998), we calculated corrected $k$ values ($k'$) by subtracting the
absolute value of the linear regression slope from the constant $k = \frac{[SO_4^{2-}]}{Na^+}$ in seawater
(Table S2), i.e. $k'_{1737-2000}$= 0.02, $k'_{1737-1949}$= 0.03, and $k'_{1950-2000}$= 0.02. The $k'$ values
recalculated for the S100 core are lower than $k'$ values described by Palmer et al.
(2002), and Plummer et al. (2012) at Law Dome ($k'_{Law\ Dome}$= 0.04, with
concentrations expressed in μmol L$^{-1}$), and similar to the $k'$ value obtained by Inoue
et al. (2017) for a Mill Island coastal core ($k'_{Mill\ Island}$= 0.03, with concentrations
expressed in μmol L$^{-1}$). Wagebach et al. (1998) reported winter $k'$ of 0.02 (with
concentrations expressed in μmol L$^{-1}$) associated to airborne sea-salt particles
experiencing ssSO$_4^{2-}$ depletion in respect to seawater, with a depletion factor
($k$= $k$/$k'$) of 5.5 for a firn core drilled at eastern Ronne Ice Shelf. The S100 core
presents depletion factors of two for the period 1737–1949, and three for the period
1950–2000.
The annual nssSO$_4^{2-}$ fraction, without the effect of sulfate fractionation, was then
recalculated using the values for $k'$ of 0.02 and 0.03 (Table 8, and Figure 4a y b, red
and blue lines).
**4    Discussion**
From the spatial and temporal variability of sea-salt concentrations in the different
FIS cores discussed here, it seems that more than one mechanism is contributing
to the load of sea-salts at FIS, in agreement with the findings by Abram et al. (2013).
The ice core data from S100 also suggest that there was a change in sea-salt
deposition regime after the 1950s evidenced by an increase, up to six-fold, of
median sea-salt concentrations after the 1950s in comparison with the previous 200
years. Although a negative trend in SMB has been observed in the S100 and KC
cores for the second half of the 20$^{th}$ century (Figure 2e and f) (Vega et al., 2016),
the 0.2 % m w.e. y$^{-1}$ decrease in accumulation registered in the S100 core after
1950 (Table S4) cannot account for the increase observed in sea-salt
concentrations after 1950s. This increase in concentration is accompanied by a
clear shift in nssSO$_4^{2-}$ toward negative values, indicative of ssSO$_4^{2-}$ depletion in
sea-salts measured in the core in comparison to bulk seawater, with ssSO$_4^{2-}$

depletion factors of two for the period 1737–1949, and three for the period 1950–2000.

The negative $nssSO_4^{2-}$ values found in the FIS records could be explained by an enhanced input of sea-salts from (i) windblown frost flowers and/or (ii) aerosol formed after fractionated salty-snow sublimation, with both (i) and (ii) being formed in the neighbouring waters at the eastern flank of FIS. Yang et al. (2008) have reported that aerosol production via (ii) can be more than one-fold larger per unit area than sea-salt production from the open ocean. There is no or very limited amount of multi-annual sea-ice near FIS, and young sea-ice formed during winter in the vicinity of the S100 site is quickly covered by snow due to cyclonic activity. Trajectory studies of air with high sea-salts concentrations and low $SO_4^{2-}/Na^+$ ratios arriving at Halley station, showed that these air masses mainly originate at regions where young sea-ice and frost flowers are formed (Hall and Wolff, 1998; Rankin and Wolff, 2002). However, conditions at Halley are not comparable to FIS, since the main easterly or north-northeasterly wind direction prevailing at Halley means an off-land air flow, thus creation of polynyas with open water and consecutive new ice formation, whereas at FIS, and most of the Dronning Maud Land coast, the wind is mainly parallel to the coast or even slightly towards the coast. In particular, a quantification of the areas covered by frost flowers is still missing. It is possible that those areas are comparatively small due to the generally high wind speeds prevailing above the Southern Ocean, resulting in a high percentage of frazil ice, and synoptic conditions lead to the quick development of a snow cover on the young sea-ice. Although it is not possible to apportion the contribution of fractionated sea-salts via (i) or (ii) with the current data, it is plausible that a larger contribution of fractionated aerosol formed from salty-snow than by frost flowers, based on recent experimental evidence that frost flowers would not be a direct source of sea-salt aerosols (Yang et al., 2017). In addition, frequent stormy conditions in the area are detrimental for the formation of frost flowers, which form under quiet, undisturbed conditions, usually only in leads or small polynyas under the influence of anticyclonic weather. This also means low wind speeds and thus not much transport of frost flowers to the sampling sites at FIS. Thus, mechanism (ii), blowing salty snow

formed on thin sea-ice that sublimates during transport to form sea-salt aerosols, appears as a much more probable explanation considering the local meteorological conditions in the study area.

Considering that we found no correlation between ion concentrations and site elevation (section 3.4), a decrease in wind transport efficiency of frost flowers (size of 10–20 mm) and aerosol formed via (ii) (size >0.95 $\mu$m) (Seguin et al., 2014) due to increased elevation cannot be addressed to explain the lower sea-salt values observed at the ice-rises compared to the S100 site. As mentioned in section 3.4, local effects on annual SMB due to topography and meteorology at the KM and BI sites, reported by Vega et al. (2016), are most likely involved in the different load of sea-salt to these sites.

The dramatic increase in fractionated sea-salt in the S100 core after 1950s could be associated with an enhanced exposure of the S100 site to primary aerosol, in addition to and enhanced production of fractionated aerosol, evidenced by a dominance of negative $nssSO_4^{2-}$ values after 1950. Figure 2 a and c show that sea-salts started to increase after 1950 with a marked peak corresponding to year 1966 (±3 years). According to Rignot et al. (2011), ice velocities near S100 were in the order of 10s–100s m y$^{-1}$ for the period 2007–2009. We hypothesize that the increase observed in sea-salts from 1950 could be linked to an increase in ice velocities in comparison to the 1737–1949 period; and that the calving event occurred at Trolltunga in 1967 (Vinje, 1975) (Figure 1), enhanced the input of fractionated sea-salts to the S100 core by modifying the sea-ice conditions around S100, leading to the marked peak found in sea-salts in 1966 (±3 years). This could be supported by the fact that negative $nssSO_4^{2-}$ values slowly decreased between 1950–1966, showing a marked minimum around 1966 (±3 years) (Figure 4 b), which could have been caused by the Trolltunga calving event. The longer Trolltunga present before the calving event could have formed a larger bay to the east of it, where compaction of the sea ice occurred due to prevailing easterly winds, resulting in thicker, longer-lasting sea-ice, which limited the sea-spray formation. Such thick sea-ice does not form under post-calving event conditions, e.g. with a shorter tongue. Post-calving event conditions would mean that more sea-spray could be

formed and deposited at the FIS sites compared with pre-calving periods. However, sea-spray enhancing alone cannot account for either the increase of sea-salt concentrations or the negative $nssSO_4^{2-}$ found in snow and ice samples. In order to explain the fractionated sea-salt values detected in the S100 cores, there must be an enhanced source of fractionated sea-salts after the calving event. This would be the case if young sea-ice (where fractionation of sea-salts can take place) formed nearby the S100 site as a result of the greater area of open sea available after the calving event. Thicker, long-lasting sea-ice present before the calving event would have been a more stable substrate, prone to less flooding through cracks and leads and most likely will present a reduced snow salinity in comparison to young sea-ice (Massom et al., 2001). Following the same supposition as Rhodes et al. (2017), i.e. that young sea-ice would be more saline than multi-year ice, it can be expected that sea-salt aerosols produced by blowing snow over sea-ice would have higher sea-salt concentrations when young-ice is formed than when multi-year sea-ice is formed, in coherence with the proposed hypothesis. The higher sea-salt concentrations found in S100 after the Trolltunga detachment, could be explained by an enhanced contribution of sea-salt aerosols entrained by blowing salty snow found over young sea-ice formed near the S100 site. If the air is unsaturated, water in these snow particles will sublimate producing fractionated sea-salt aerosol. As schematized in Figure 2 in Rhodes et al. (2017), the sea-salt aerosol can be transported inland and be deposited either by dry or wet deposition. Since sea-salt concentrations are much higher at the S100 core than the ice-rises cores, it is plausible that most of the flux of sea-salts at the S100 site is due to dry deposition, due to the short distance from the coast and low elevation, while deposition at the ice-rises would be balanced between the wet and dry regimes. Rhodes et al. (2017) found a marked gradient in the sea-ice sea-salts to oceanic sea-salts (produced by bubble bursting) ratio in Arctic sites, whit higher ratios closer to the sea ice source and when the location is in the path between sea-ice and prevailing winds. To test the hypothesis presented here, a closer analysis of satellite and historical sea-ice data and a model-based study to estimate the spatial and elevation gradient of sea-

ice sea-salts to FIS can be done, which, however, is beyond the scope of the present
study.
Other possible mechanisms, such as deposition of sea-salts with rime or windblown
snow present over multi-annual sea ice, can explain neither the increase in sea salt
concentration nor the fractionation observed in S100 after the 1950s. Additionally,
annual averages of monthly zonal and meridional wind speeds (ERA40, Uppala et
al., 2005) for the area (69°S–71°S, 3.5°W–5°E) between 1955–2001 (Figure 6)
show no significant positive trends, thus evidencing that the S100 sea-salt increase
after 1950s cannot be related to enhanced transport by wind.
Due to the limited time coverage of the KC, KM, and BI cores, we do not know
whether there was a relative increase in sea-salt concentrations in the ice-rises
cores after the 1950s influenced by the Trolltunga calving. Due to the large input of
bio-nssSO$_4^{2-}$ to the ice-rises sites, any possible signal of fractionated sea-salts in
any of the ice-rises cores could be easily masked by the biogenic fraction (e.g. no
significant negative nssSO$_4^{2-}$ values would be observed). Relatively higher sea-salt
concentrations measured in the KM core in comparison to the other ice-rises cores
could be explained by a combination of distance to the sea and the prevailing
precipitation and wind conditions in the area: precipitation on FIS is mainly caused
by frontal systems of cyclones in the circumpolar trough that move eastwards north
of the coast, thus leading to easterly or east-north-easterly surface winds on FIS
(Schlosser et al., 2008). This means that even though BI is equally close to the sea
as KM (Figure 1), KM has by far the shortest distance to the source of marine
aerosols of all three cores, which could explain the comparatively high sea-salt
concentrations (Table 1).
**5   Conclusions**
This study reports sub-annual and long-term temporal sea-salt and major ion
concentration changes measured in three recently drilled firn cores from different
ice-rises located at Fimbul Ice Shelf (FIS): Kupol Ciolkovskogo, Kupol Moskovskij,
and Blåskimen Island, and a 100 m long core drilled near the FIS edge (S100). No
significant relationship is found between the median annual ion concentrations and
latitude, site elevation, and distance from the sea for any of the species, and only

annual $SO_4^{2-}$ and MSA concentrations show a significant decrease (at the 95 % confidence level) with latitude and east longitude, respectively. A significant increase in sea-salts is observed in the S100 core after the 1950s, which is associated with an enhanced exposure of the S100 site to primary sea-salt aerosol, and enhanced input of fractionated sea-salts. This increase in sea-salt concentrations was accompanied by a shift in $nssSO_4^{2-}$ toward negative values, suggesting the input of fractionated sea-salts to the ion load in the S100 core most likely by enhancing sea-salts production by blowing salty snow over sea-ice. Due to the large input of bio-$nssSO_4^{2-}$ to the ice-rises cores, it is hard to assess the degree of $ssSO_4^{2-}$ depletion in snow in comparison to bulk seawater at these sites. Consequently, the results of this study suggest that the S100 record contains a sea-salt record dominated by processes of sea-ice formation in the neighbouring waters, whereas the ice-rises cores record the signal of larger-scale conditions of atmospheric flow, large inputs of bio-$nssSO_4^{2-}$, and less efficient transport of sea-salts evidenced by lower mean concentrations in comparison to the S100 site. These findings are of vital importance for the understanding of the mechanisms of sea-salt aerosol production, transport and deposition at coastal Antarctic sites, and for the improvement of the current Antarctic sea-ice reconstructions based on sea-salt chemical proxies.

## 6   Data availability

For the chemistry profiles of the KC, KM, BI, and S100 cores, and FIS snow pits, please contact E. Isaksson (elisabeth.isaksson@npolar.no).

MODIS Mosaic of Antarctica (MOA) image is available through the GIS package Quantarctica, version 2.0 at http://quantarctica.npolar.no/.

ERA40 reanalysis data is available at https://climatedataguide.ucar.edu/climate-data/era40 (Uppala, et al., 2005).

**Acknowledgements**

We are grateful to those who helped to collect, transport, sample and analyse the firn cores and snow pits at FIS. We would like to thank V. Goel and J. van Oostveen

for providing the 50-m contours and the pre-calving extent of Trolltunga,
respectively, used in Figure 1, and T. Maldonado for processing the data for
Figure 6. We thank the Norwegian Polar Institute´s team behind the Quantarctica
package. Financial support came from Norwegian Research Council through NARE
and the Centre for Ice, Climate and Ecosystems (ICE) at the Norwegian Polar
Institute in Tromsø. Additional support was received from University of Costa Rica,
network ISONet (project B6-774).

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

**Tables**
Table 1. Cores (KC, KM, BI, S100) locations and sampling details. Distances of the
core locations to the ice shelf side were obtained using the GIS package
Quantarctica (www.quantarctica.org). (*) refers to Kaczmarska et al. (2004), and (§)
to Vega et al. (2016).

| Site | Location | Elevation | Core length *Ice depth* Ice temp. at 10 m | Distance from the coast | Time coverage | Average SMB |
|------|----------|-----------|-------------------------------------------|-------------------------|---------------|-------------|
|      |          | (m a.s.l.) | (m) | (km) | (years) | (m w.e. y$^{-1}$) |
| KC[§] | 70°31′S, 2°57′E | 264 | 20.0 *460* −17.5 | 42 | (1958–2012) ±3 | 0.24 |
| KM[§] | 70°8′S, 1°12′E | 268 | 19.6 *410* −15.9 | 12 | (1995–2014) ±1 | 0.68 |
| BI[§] | 70°24′S, 3°2′W | 394 | 19.5 *460* −16.1 | 10 | (1996–2014) ±1 | 0.70 |
| S100* | 70°14′S, 4°48′E | 48 | 100 - −17.5 | 3 | (1737–2000) ±3 | 0.30 |

Table 2. Median, mean, maximum, minimum, and standard deviation ($\sigma$) of ion
concentrations (in $\mu$mol L$^{-1}$) in the KC, KM, BI, and S100 firn/ice cores. Ion
concentrations at the top 2 m of the KC, KM, and BI cores were not measured. Non-
detected concentrations were set as half the detection limit of each ion. Note: (*) the
period is 1958.5–2012, (-) not measured. Values of water stable isotopes and
deuterium excess for the KC, KM, and BI are reported by Vega et al. (2016).

| Site | Period (years) | MSA | Cl$^-$ | NO$_3^-$ | SO$_4^{2-}$ | Na$^+$ | K$^+$ | Mg$^{2+}$ | Ca$^{2+}$ |
|------|------|------|------|------|------|------|------|------|------|
| | | | | | Median Mean Maximum Minimum $\sigma$ ($\mu$mol L$^{-1}$) | | | | |
| KC | 1958–2007 | 0.2 | 10.0 | 0.6 | 1.8 | 9.4 | 0.2 | 0.9 | 0.5 |
| | | 0.2 | 11.3 | 0.7 | 2.1 | 12.1 | 0.2 | 1.0 | 1.8 |
| | | 0.9 | 59.3 | 1.8 | 10.3 | 162.6 | 1.5 | 4.2 | 62.7 |
| | | 0.0 | 1.7 | 0.2 | 0.1 | 1.1 | 0.0 | 0.2 | 0.2 |
| | | 0.2 | 7.0 | 0.4 | 1.5 | 12.2 | 0.2 | 0.6 | 6.5 |
| KM | 1995–2012 | 0.3 | 71.3 | 0.4 | 4.5 | 57.7 | 1.5 | 6.3 | 1.6 |
| | | 0.5 | 119.7 | 0.5 | 6.2 | 88.6 | 2.0 | 8.8 | 2.2 |
| | | 9.4 | 571.6 | 5.4 | 84.5 | 654.8 | 16.1 | 45.5 | 10.6 |
| | | 0.0 | 6.9 | 0.1 | 0.5 | 2.9 | 0.1 | 1.0 | 0.4 |
| | | 0.6 | 104.4 | 0.4 | 7.5 | 92.6 | 1.8 | 7.4 | 1.8 |
| BI | 1996–2012 | 0.4 | 23.1 | 0.4 | 1.9 | 19.0 | 0.5 | 2.0 | 0.6 |
| | | 0.5 | 27.0 | 0.5 | 2.5 | 22.5 | 0.6 | 2.4 | 0.7 |
| | | 2.0 | 185.8 | 2.3 | 11.2 | 161.8 | 5.0 | 15.9 | 3.5 |
| | | 0.0 | 1.8 | 0.1 | 0.3 | 1.7 | 0.0 | 0.3 | 0.3 |
| | | 0.4 | 20.3 | 0.4 | 1.9 | 17.3 | 0.5 | 1.7 | 0.4 |
| S100 | 1737–2000 | 0.1 | 20.9 | 0.5 | 1.2 | 20.7 | 0.4 | 2.0 | 0.7 |
| | | 0.2 | 78.2 | 0.6 | 2.6 | 75.5 | 1.6 | 4.4 | 1.8 |
| | | 5.6 | 2174.1 | 1.8 | 56.0 | 1315.5 | 39.6 | 35.9 | 40.0 |
| | | 0.0 | 3.7 | 0.1 | 0.2 | 3.7 | 0.1 | 0.0 | 0.1 |
| | | 0.3 | 187.7 | 0.3 | 5.2 | 149.8 | 3.4 | 5.4 | 3.6 |
| S100 | 1995–2000 | 0.1 | 132.4 | 0.6 | 3.2 | 144.0 | 3.3 | 10.7 | 3.0 |
| | | 0.2 | 220.8 | 0.6 | 6.0 | 209.0 | 4.4 | 10.8 | 4.2 |
| | | 1.0 | 2174.1 | 1.4 | 35.8 | 1315.5 | 39.6 | 35.9 | 40.0 |
| | | 0.0 | 11.3 | 0.1 | 0.8 | 11.0 | 0.3 | 1.5 | 0.5 |
| | | 0.2 | 332.1 | 0.3 | 7.4 | 232.0 | 5.9 | 6.2 | 6.0 |
| S100 | 1737–1949 | 0.1 | 16.0 | 0.6 | 1.0 | 15.1 | 0.3 | 1.4 | 0.5 |
| | | 0.1 | 18.8 | 0.6 | 1.1 | 18.4 | 0.4 | 1.6 | 0.6 |
| | | 0.8 | 120.9 | 1.8 | 4.8 | 138.8 | 2.1 | 6.3 | 5.7 |
| | | 0.0 | 3.7 | 0.1 | 0.2 | 3.7 | 0.1 | 0.3 | 0.1 |
| | | 0.1 | 12.9 | 0.3 | 0.5 | 14.2 | 0.2 | 0.9 | 0.5 |
| S100 | 1950–2000 | 0.1 | 88.5 | 0.5 | 2.8 | 98.2 | 2.0 | 7.9 | 1.9 |
| | | 0.2 | 179.2 | 0.6 | 5.2 | 172.6 | 3.6 | 9.1 | 3.7 |
| | | 5.6 | 2174.1 | 1.5 | 56.0 | 1315.5 | 39.6 | 35.9 | 40.0 |
| | | 0.0 | 9.1 | 0.1 | 0.4 | 8.6 | 0.2 | 0.0 | 0.3 |
| | | 0.5 | 280.9 | 0.3 | 7.9 | 213.2 | 5.0 | 6.4 | 5.4 |

Table 3. PCA loadings of the first three (KC) and two (KM, BI, and S100) principal components calculated at an annual resolution in a set of 8 different ions measured in the ice-rises and S100 cores. PCA loadings were obtained at three different time intervals in the S100 core: 1737–2000, 1737–1949, and 1950–2000. Sources related to the different components are displayed in the bottom row.

| Core | KC | | | KM | | BI | | S100 | | | | | |
| --- | --- | --- | --- | --- | --- | --- | --- | --- | --- | --- | --- | --- | --- |
| Resolution | Annual | | | Annual | | annual | | annual (1737–2000) | | annual (1737–1949) | | annual (1950–2000) | |
| Loadings | PC1 | PC2 | PC3 | PC1 | PC2 | PC1 | PC2 | PC1 | PC2 | PC1 | PC2 | PC1 | PC2 |
| MSA | 0.17 | **0.52** | -0.19 | -0.20 | **0.64** | 0.03 | **0.65** | 0.16 | **0.54** | 0.23 | **0.44** | 0.03 | **0.73** |
| Cl | **0.46** | -0.17 | -0.19 | **0.40** | 0.03 | **0.43** | -0.07 | **0.42** | -0.07 | **0.43** | -0.11 | **0.42** | -0.08 |
| NO₃⁻ | 0.13 | **0.59** | 0.35 | -0.26 | **0.56** | -0.03 | **0.56** | -0.06 | **0.79** | -0.08 | **0.74** | 0.14 | **0.60** |
| SO₄²⁻ | 0.33 | **0.47** | 0.08 | 0.30 | **0.50** | 0.30 | **0.48** | **0.37** | 0.23 | 0.30 | **0.45** | **0.38** | 0.23 |
| Na⁺ | **0.44** | -0.11 | -0.22 | **0.40** | 0.07 | **0.43** | -0.06 | **0.42** | -0.09 | **0.43** | -0.13 | **0.42** | -0.10 |
| K⁺ | **0.46** | -0.19 | -0.11 | **0.40** | 0.03 | **0.40** | -0.10 | **0.41** | -0.06 | **0.41** | -0.10 | **0.42** | -0.05 |
| Mg²⁺ | **0.45** | -0.15 | 0.11 | **0.39** | 0.08 | **0.43** | -0.08 | **0.41** | -0.10 | **0.41** | -0.11 | **0.40** | -0.17 |
| Ca²⁺ | 0.17 | -0.24 | **0.85** | **0.40** | 0.10 | **0.43** | -0.03 | **0.39** | 0.02 | **0.36** | 0.05 | **0.39** | -0.07 |
| Explained Variance (%) | 51 | 22 | 12 | 76 | 18 | 65 | 24 | 70 | 15 | 60 | 17 | 69 | 16 |
| Source | sea-salts | biogenic mixed | terrestrial | sea-salts terrestrial | biogenic mixed | sea-salts terrestrial | biogenic mixed | sea-salts terrestrial | biogenic mixed | sea-salts terrestrial | biogenic mixed | sea-salts terrestrial | biogenic mixed |

Table 4. Correlation coefficients (R) for the median annual ion concentrations at the
different cores (KC, KM, BI, and S100) vs. latitude, longitude, site elevation, and
distance from the ice shelf edge, for the overlapping period 1997–2000. Significant
values at the 95 % confidence interval are shown in bold.

| R | MSA | $Cl^-$ | $NO_3^-$ | $SO_4^{2-}$ | $Na^+$ | $K^+$ | $Mg^{2+}$ | $Ca^{2+}$ |
|---|---|---|---|---|---|---|---|---|
| Latitude (˚S) | 0.20 | −0.84 | 0.89 | **−0.98** | −0.84 | −0.78 | −0.91 | −0.81 |
| Longitude (˚W) | **−0.99** | 0.6 | 0.15 | 0.28 | 0.58 | 0.60 | 0.51 | 0.60 |
| Elevation (m a.s.l.) | 0.94 | −0.81 | −0.04 | −0.41 | −0.81 | −0.85 | −0.73 | −0.84 |
| Distance from the sea (km) | −0.06 | −0.70 | 0.54 | −0.59 | −0.70 | −0.71 | −0.72 | −0.69 |

Table 5. Cl⁻/Na⁺ ratio (expressed in $\mu$mol L$^{-1}$), nssNa⁺, ssNa⁺, and percentage of nssNa⁺ to total Na⁺ in the KC, KM, BI, and S100 cores. Since some of the calculated ssNa⁺ values in the KC core were negative, ssNa⁺ statistics are shown considering all data points, and only positive ssNa⁺ values (sample rejection percentage is shown in parenthesis).

| Site | Period (years) | Ratio Cl⁻/Na⁺ | nssNa⁺ (crustal) | ssNa⁺ All values | ssNa⁺ Only positive values | nssNa⁺ to total Na⁺ (%) |
|---|---|---|---|---|---|---|
| | | | | Median Mean Maximum Minimum $\sigma$ ($\mu$mol L$^{-1}$) | | |
| KC | 1958–2007 | 1.2 1.1 1.9 0.1 0.3 | 0.7 2.6 87.7 0.3 9.1 | 8.1 9.6 159.1 -67.3 14.5 | (5 %) 8.4 11.2 159.1 0.4 12.1 | 21 |
| KM | 1995–2012 | 1.3 1.3 3.8 0.8 0.2 | 2.2 3.0 14.8 0.5 2.5 | 54.9 85.6 644.4 2.4 90.2 | - | 3 |
| BI | 1996–2012 | 1.2 1.2 1.5 0.9 0.1 | 0.8 1.0 4.7 0.4 0.5 | 18.1 21.5 156.8 1.3 16.8 | - | 4 |
| S100 | 1737–2000 | 1.0 1.0 2.1 0.1 0.2 | 1.0 2.5 56.0 0.1 5.1 | 19.5 73.0 1259.4 3.5 145.16 | - | 5 |
| S100 | 1995–2000 | 1.0 1.0 2.1 0.1 0.2 | 9.1 9.2 30.5 1.3 5.2 | 135.3 199.8 1285.0 8.6 227.1 | - | 4 |
| S100 | 1737–1949 | 1.0 1.1 1.8 0.6 0.2 | 1.2 1.4 5.3 0.2 0.8 | 13.7 17.1 138.0 3.2 13.7 | - | 8 |
| S100 | 1950–2000 | 1.0 1.0 2.1 0.1 0.2 | 6.7 7.7 30.5 0.0 5.6 | 90.5 164.9 1285.0 6.4 208.2 | - | 4 |

Table 6. Median, mean, maximum, minimum, and standard deviation of ss- and nss-fractions in the KC, KM, BI, and S100 cores. Percentage of negative nss-values for each ion is shown in parenthesis. Negative ss-values in the KC core are due to the 5% of $ssNa^+$ negative values obtained in section 3.5 (Table 5).

| Core | Period (years) | Cl⁻ ss | Cl⁻ nss | SO₄²⁻ ss | SO₄²⁻ nss | K⁺ ss | K⁺ nss | Mg²⁺ ss | Mg²⁺ nss |
|---|---|---|---|---|---|---|---|---|---|
| | | | Median Mean Maximum Minimum σ (μmol L⁻¹) | | | | | | |
| KC | 1958–2007 | | (28 %) | | (3 %) | | (27 %) | | (41 %) |
| | | 9.4 | 0.9 | 0.5 | 1.2 | 0.2 | 0.0 | 0.9 | 0.1 |
| | | 11.1 | 0.2 | 0.6 | 1.6 | 0.2 | 0.1 | 1.1 | 0.0 |
| | | 184.6 | 96.6 | 9.6 | 8.2 | 3.2 | 1.8 | 17.5 | 9.5 |
| | | −78.6 | −165.6 | −4.0 | −0.5 | −1.4 | −2.0 | −7.4 | −16.8 |
| | | 16.8 | 15.4 | 0.9 | 1.4 | 0.3 | 0.2 | 1.6 | 1.6 |
| KM | 1995–2012 | | (5 %) | | (38 %) | | (12 %) | | (43 %) |
| | | 63.8 | 6.2 | 3.3 | 0.9 | 1.1 | 0.2 | 6.1 | 0.2 |
| | | 99.3 | 10.4 | 5.1 | 1.1 | 1.7 | 0.3 | 9.4 | −0.7 |
| | | 747.5 | 80.5 | 38.7 | 45.9 | 12.9 | 13.8 | 70.9 | 23.6 |
| | | 2.8 | −200.3 | 0.1 | −15.1 | 0.1 | m8.4 | 0.3 | −54.1 |
| | | 104.7 | 19.4 | 5.4 | 4.5 | 1.8 | 1.3 | 9.9 | 6.7 |
| BI | 1996–2012 | | (4 %) | | (26 %) | | (2 %) | | (42 %) |
| | | 21.0 | 1.7 | 1.1 | 0.8 | 0.4 | 0.1 | 2.0 | 0.0 |
| | | 25.0 | 2.1 | 1.3 | 1.3 | 0.4 | 0.1 | 2.4 | 0.0 |
| | | 181.9 | 15.3 | 9.4 | 7.4 | 3.1 | 4.5 | 17.3 | 2.0 |
| | | 1.6 | −1.7 | 0.1 | −1.4 | 0.0 | 0.0 | 0.2 | −3.0 |
| | | 19.5 | 1.8 | 1.0 | 1.7 | 0.3 | 0.3 | 1.9 | 0.5 |
| S100 | 1737–2000 | | (81 %) | | (51 %) | | (40 %) | | (74 %) |
| | | 22.7 | −2.1 | 1.2 | 0.0 | 0.4 | 0.0 | 2.2 | −0.3 |
| | | 84.7 | −6.6 | 4.4 | −1.7 | 1.5 | 0.1 | 8.0 | −3.7 |
| | | 1460.9 | 713.2 | 75.6 | 5.2 | 25.2 | 14.4 | 138.5 | 3.7 |
| | | 4.0 | −583.3 | 0.2 | −44.6 | 0.1 | −2.7 | 0.4 | −102.7 |
| | | 168.4 | 54.2 | 8.7 | 5.0 | 2.9 | 0.9 | 16.0 | 11.3 |
| S100 | 1995–2000 | | (87 %) | | (78 %) | | (39 %) | | (74 %) |
| | | 162.0 | −23.7 | 8.4 | −4.5 | 2.8 | 0.1 | 15.4 | −5.0 |
| | | 235.6 | −14.8 | 12.2 | −6.2 | 4.1 | 0.3 | 22.3 | −11.5 |
| | | 1460.9 | 713.2 | 75.6 | 3.2 | 25.2 | 14.4 | 138.5 | 3.5 |
| | | 11.3 | −583.3 | 0.6 | −39.8 | 0.2 | −1.2 | 1.1 | −102.7 |
| | | 259.9 | 134.6 | 13.4 | 8.2 | 4.5 | 2.1 | 24.6 | 18.9 |
| S100 | 1737–1949 | | (75 %) | | (33 %) | | (41 %) | | (73 %) |
| | | 16.6 | −1.1 | 0.9 | 0.2 | 0.3 | 0.0 | 1.6 | −0.2 |
| | | 20.4 | −1.6 | 1.1 | 0.1 | 0.4 | 0.0 | 1.9 | −0.3 |
| | | 159.4 | 10.4 | 8.3 | 4.1 | 2.8 | 0.6 | 15.1 | 3.7 |
| | | 4.0 | −44.5 | 0.2 | −7.2 | 0.1 | −0.6 | 0.4 | −14.1 |
| | | 16.1 | 4.5 | 0.8 | 0.8 | 0.3 | 0.1 | 1.5 | 1.2 |
| S100 | 1950–2000 | | (90 %) | | (81 %) | | (38 %) | | (74 %) |
| | | 108.6 | −14.7 | 5.6 | −2.7 | 1.9 | 0.1 | 10.3 | −2.3 |
| | | 194.2 | −15.0 | 10.1 | −4.8 | 3.4 | 0.2 | 18.4 | −9.3 |
| | | 1460.9 | 713.2 | 75.6 | 5.2 | 25.2 | 14.4 | 138.5 | 3.5 |
| | | 9.1 | −583.3 | 0.5 | −44.6 | 0.2 | −2.7 | 0.9 | −102.7 |
| | | 239.3 | 88.5 | 12.4 | 7.1 | 4.1 | 1.4 | 22.7 | 17.0 |

Table 7. Median, mean, minimum, maximum, and standard deviation ($\sigma$) of MSA/nssSO$_4^{2-}$ ratios, and bio-nssSO$_4^{2-}$ in the KC, KM, BI, and S100 cores. Statistics for the MSA/nssSO$_4^{2-}$ ratio are presented considering all values, and only positive values (sample rejection percentage is shown in parenthesis). In addition, the percentage of bio-nssSO$_4^{2-}$ to total SO$_4^{2-}$ is shown for all the cores.

| Site | Period (years) | Statistic | MSA/nssSO$_4^{2-}$ All values | MSA/nssSO$_4^{2-}$ Only positive values | bio-nssSO$_4^{2-}$ | bio-nssSO$_4^{2-}$ to total SO$_4^{2-}$ (%) |
|---|---|---|---|---|---|---|
| KC | 1958–2007 | (3 %) | | | | |
| | | Median | 0.1 | 0.1 | 1.0 | |
| | | Mean | 0.3 | 0.4 | 1.2 | 58 |
| | | Maximum | −17.2 | 14.9 | 0.0 | |
| | | Minimum | 14.9 | 0.0 | 5.2 | |
| | | $\sigma$ | 1.8 | 1.4 | 1.0 | |
| KM | 1995–2012 | (38 %) | | | | |
| | | Median | 0.1 | 0.3 | 1.9 | |
| | | Mean | 0.7 | 1.7 | 2.9 | 136 |
| | | Maximum | −12.9 | 138.8 | 0.1 | |
| | | Minimum | 138.8 | 0.0 | 52.3 | |
| | | $\sigma$ | 9.9 | 12.4 | 3.5 | |
| BI | 1996–2012 | (25 %) | | | | |
| | | Median | 0.3 | 0.3 | 2.1 | |
| | | Mean | −0.4 | 0.7 | 2.7 | 108 |
| | | Maximum | −245.1 | 30.6 | 0.1 | |
| | | Minimum | 30.6 | 0.0 | 11.3 | |
| | | $\sigma$ | 13.1 | 2.1 | 2.4 | |
| S100 | 1737–2000 | (51 %) | | | | |
| | | Median | 0.0 | 0.3 | 0.7 | |
| | | Mean | 0.2 | 1.0 | 1.0 | 37 |
| | | Maximum | −15.6 | 11.3 | 0.0 | |
| | | Minimum | 11.3 | 0.0 | 31.4 | |
| | | $\sigma$ | 1.8 | 1.9 | 1.6 | |
| S100 | 1995–2000 | (78 %) | | | | |
| | | Median | 0.0 | 0.2 | 0.5 | |
| | | Mean | 0.1 | 1.1 | 1.0 | 17 |
| | | Maximum | −1.7 | 9.5 | 0.1 | |
| | | Minimum | 9.5 | 0.1 | 5.4 | |
| | | $\sigma$ | 1.4 | 2.7 | 1.2 | |
| S100 | 1737–1949 | (33 %) | | | | |
| | | Median | 0.2 | 1.0 | 0.7 | |
| | | Mean | 0.3 | 1.0 | 0.8 | 72 |
| | | Maximum | −15.6 | 10.3 | 0.0 | |
| | | Minimum | 10.3 | 0.0 | 4.4 | |
| | | $\sigma$ | 2.0 | 1.7 | 0.6 | |
| S100 | 1950–2000 | (81 %) | | | | |
| | | Median | 0.0 | 0.2 | 0.7 | |
| | | Mean | 0.1 | 1.2 | 1.2 | 24 |
| | | Maximum | −2.6 | 11.3 | 0.1 | |
| | | Minimum | 11.3 | 0.0 | 31.4 | |
| | | $\sigma$ | 1.4 | 2.8 | 2.6 | |

1   Table 8. Median annual $nssSO_4^{2-}$ concentrations (in $\mu mol\ L^{-1}$) in the KC, KM, BI, and

2   S100 firn/ice cores. (-) Not re-calculated.

| Site | Period (years) | Median ($\mu mol\ L^{-1}$) $nssSO_4^{2-}$ $k=0.06$ | Median ($\mu mol\ L^{-1}$) $nssSO_4^{2-}$ $k'=0.02$ | Median ($\mu mol\ L^{-1}$) $nssSO_4^{2-}$ $k'=0.03$ |
|------|----------------|------|------|------|
| KC | 1958–2007 | 1.2 | - | - |
| KM | 1995–2012 | 0.9 | - | - |
| BI | 1996–2012 | 0.8 | - | - |
| S100 | 1737–2000 | 0.3 | 0.9 | 0.7 |
| S100 | 1995–2000 | −2.1 | 2.4 | 1.3 |
| S100 | 1737–1949 | 0.4 | 0.8 | 0.7 |
| S100 | 1950–2000 | −1.5 | 1.3 | 0.7 |

**Figures**

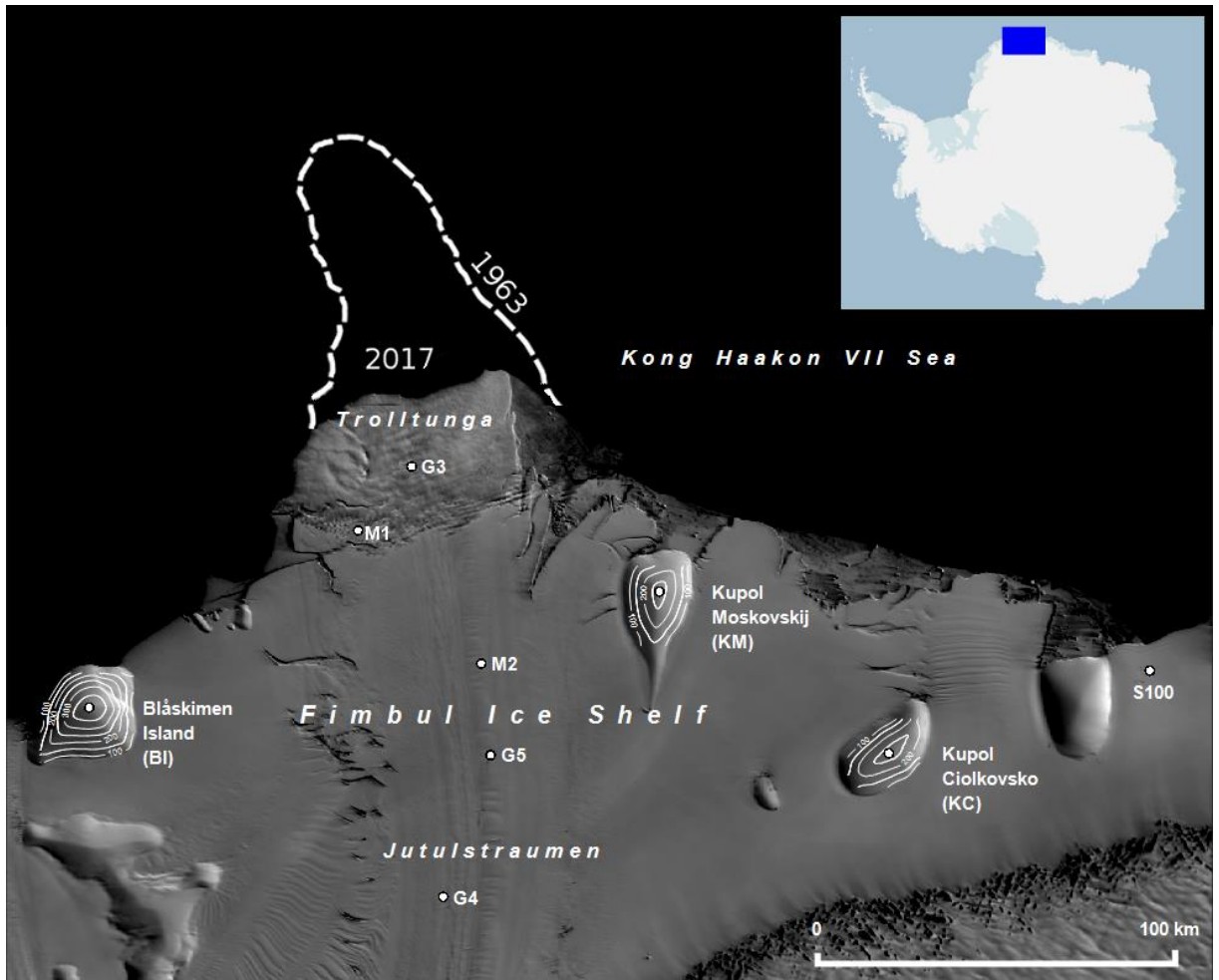

Figure 1. Satellite image of Fimbul Ice Shelf (FIS) showing the KC, KM, BI, and S100
core sites, the M1, M2, G3, G4, and G5 snow pit sites (Supplementary material),
Jutulstraumen, and Trolltunga. In addition, 50-m contours are shown at each ice-rise,
as derived from GPS profiles (V. Goel, personal communication, 2016). In addition, the
dashed line shows the extent of Trolltunga according to Corona Satellite data from
1963 (J. van Oostveen, personal communication, 2017). Map image is from the MODIS
Mosaic of Antarctica (MOA). Additional information regarding the sampling sites and
traverses in FIS can be found in Schlosser et al. (2014) and Vega et al. (2016).

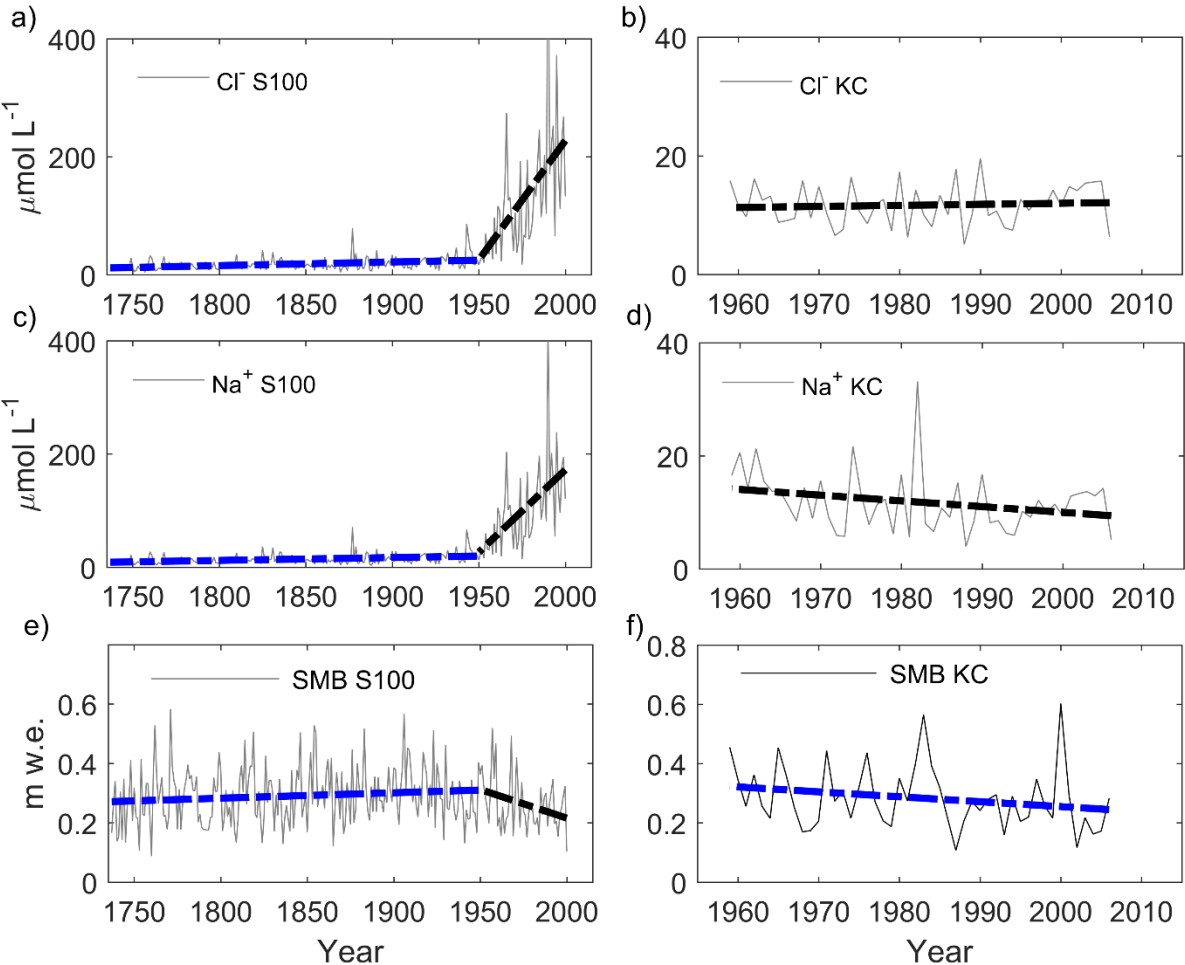

Figure 2. Annual sea-salt (Cl⁻ and Na⁺) concentrations and surface mass balance (SMB) in the two longest records retrieved at Fimbul Ice shelf, S100, (a), (c), and (e), and KC, (b), (d) and (f). Linear trends in Cl⁻ and Na⁺ concentration, and SMB measured in the S100 core are shown for two different periods: 1737–1949 (blue dashed line) and 1950–2000 (black dashed line) in (a), (c), and (e), respectively. Linear trends in Cl⁻ and Na⁺ concentrations, and SMB measured in the KC core are shown for the period 1958–2007 (black dashed line) in (b), (d) and (f), respectively. Significance, slope, and standard error of the linear regressions are given in Table S4.

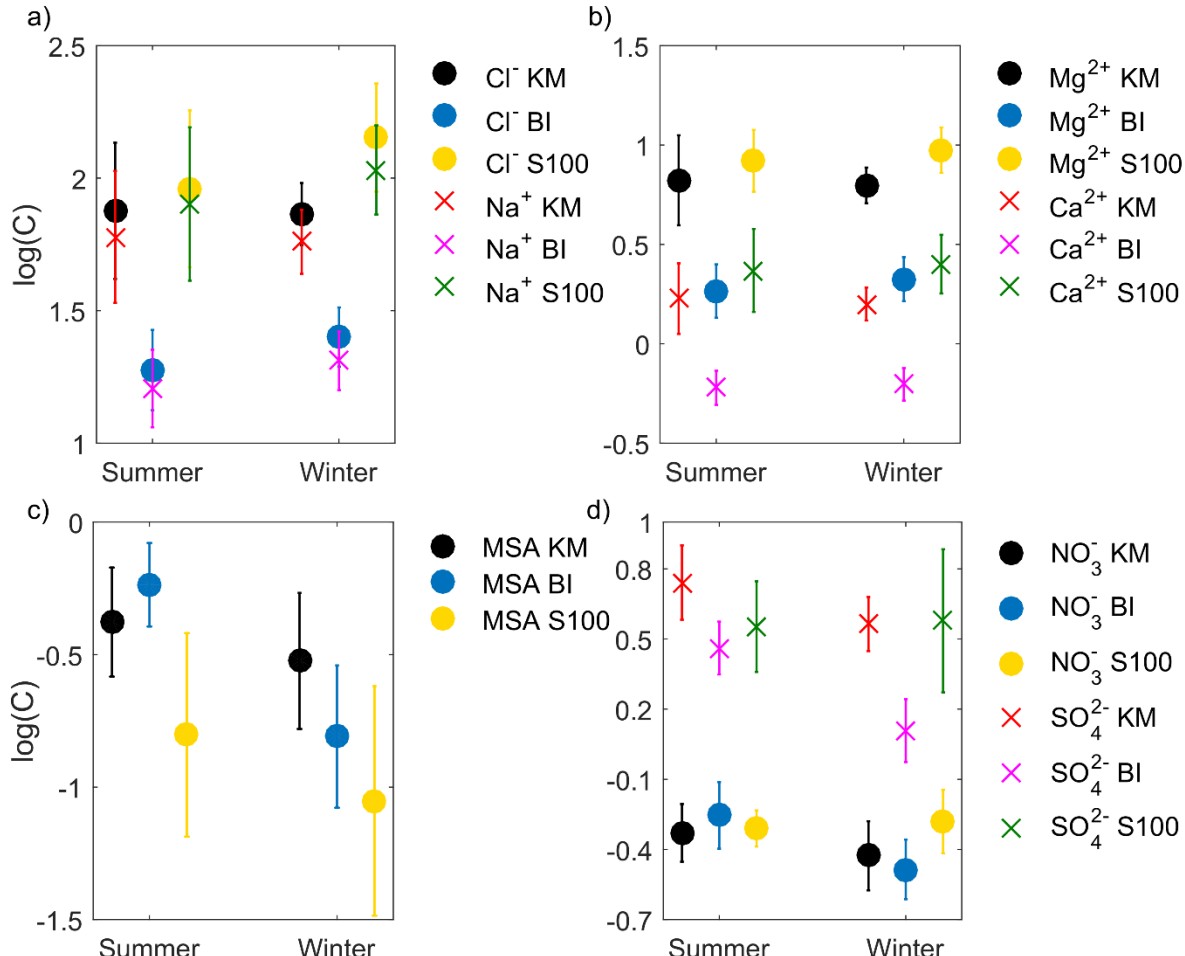

Figure 3. Sub-annual variability of selected ions, $Cl^-$ and $Na^+$ (a), $Mg^{2+}$ and $Ca^{2+}$ (b),
MSA (c) and $NO_3^-$ and $SO_4^{2-}$ (d) in cores KM, BI, and S100. Mean summer and winter
concentrations were calculated for the months NDJFMA, and MJJASO, for a period of
16, 15, and 5 years in the KM, BI, and S100 cores, respectively.

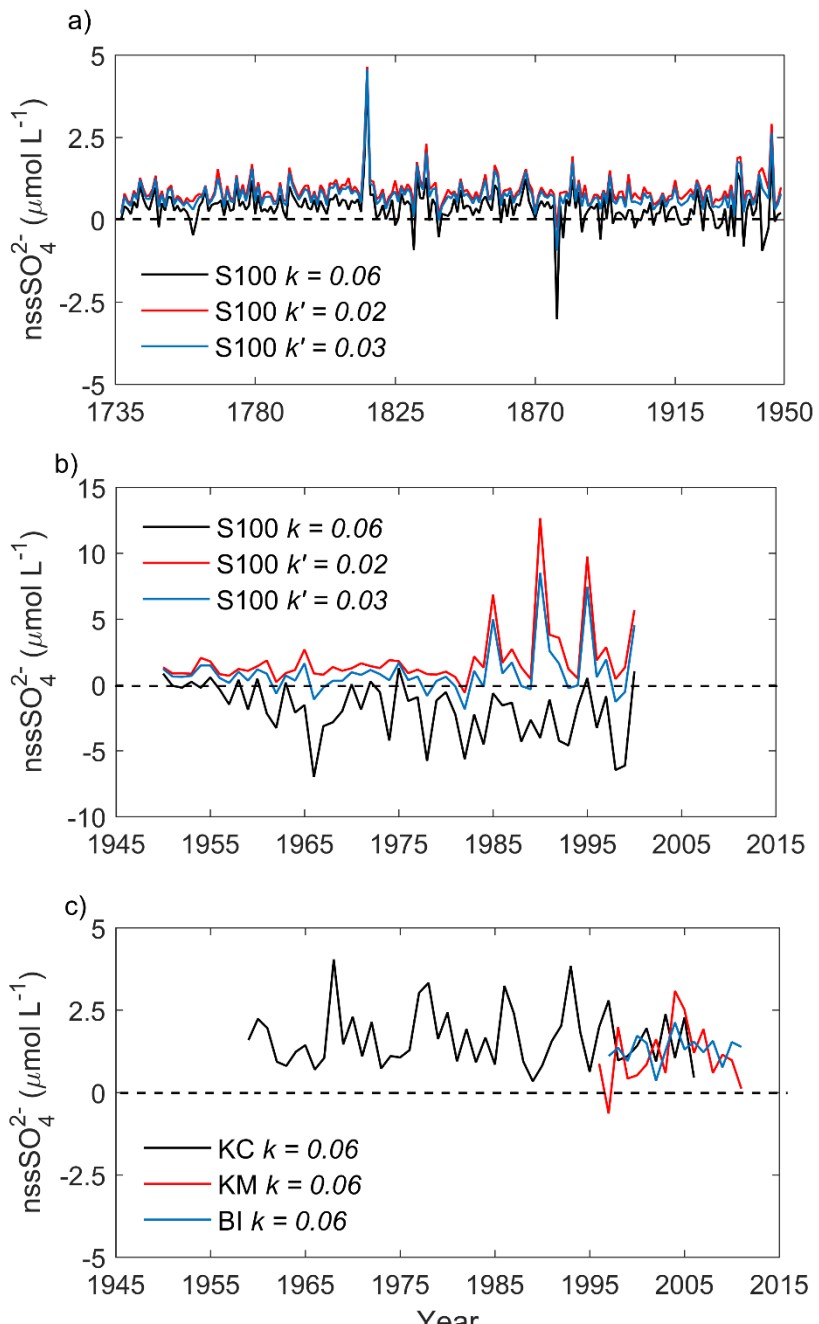

Figure 4. Annual nssSO$_4^{2-}$ concentrations in the S100 core between a) 1737–1949, b) 1950–2000, and c) in the KC, KM, and BI cores. nssSO$_4^{2-}$ recalculated using $k$= 0.06, $k'$=0.02 and $k'$=0.03 are shown in panels a) and b) with black, red and blue lines, respectively. nssSO$_4^{2-}$ in the KC, KM, and BI cores was calculated using $k$= 0.06.

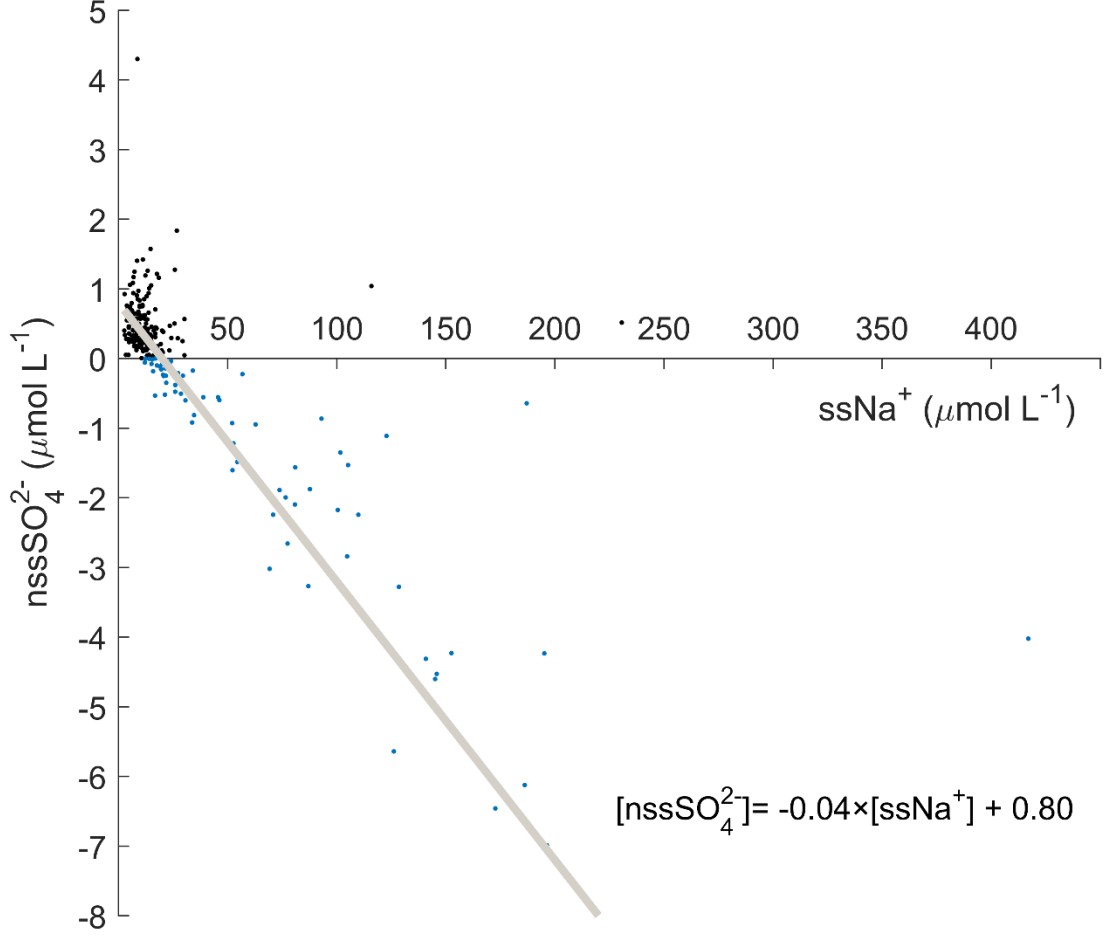

Figure 5. Scatter plot of annual $nssSO_4^{2-}$ vs. $ssNa^+$ concentrations in the S100 core.
$nssSO_4^{2-}$ was calculated using the seawater ratio as described in section 2.3 and using
a $k$=0.06 (in $\mu mol\ L^{-1}$). Positive $nssSO_4^{2-}$ values are denoted with black dots, while
negative values are denoted with blue dots. A linear regression was calculated using
all $nssSO_4^{2-}$ data points to infer corrected $k$ value ($k'$), following the approach by
Wagenbach et al. (1998).

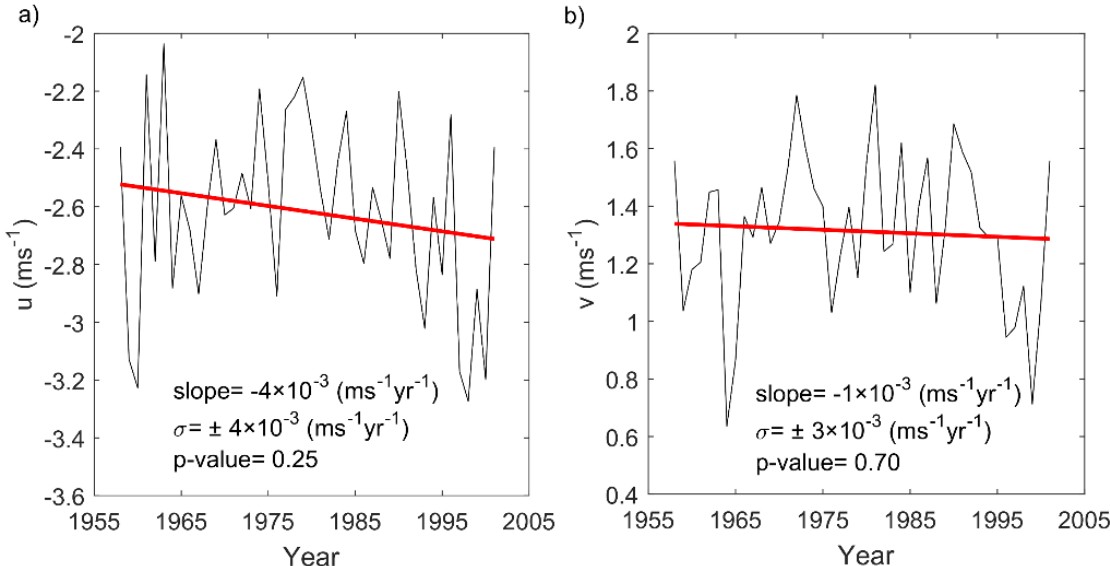

Figure 6. Annual averages of monthly a) zonal, and b) meridional wind speeds (ERA40) for the area (69°S–71°S, 3.5°W–5°E) between 1958–2001. Slope, standard deviation, and *p*-value of the linear regression are shown in the figure.