# Peer review of "Spatial and temporal variability of sea-salts in ice/firn"

_The Cryosphere, 2017_

## Referee Comment (RC1) · Anonymous Referee #1 · 21 Aug 2017

This paper presents new ice core chemical data for a coastal region of Antarctica. It interprets particularly the sea salt chemistry, and attempts to discuss the mechanisms behind sea salt production and deposition from the data. The data have some interest, particularly the unusual record from S100, and it may be possible to make a workable paper out of them. (The application of this paper is somewhat reduced because sites so close to the sea can be interesting but do not tell us too much about processes affecting inland sites.) However at the moment the paper suffers from three very major flaws, and an omission that render most of the interpretations dubious:

1. The snowpit data are all from samples covering less than a year of snowfall. This

makes it impossible to use the average values generated quantitatively, both because they are not a real yearly value, and because interannual variability means that the average for one year should have a huge uncertainty on it. There was a time when we were so desperate for new data from unexplored parts of the continent that we would at least consider surface snow data from part-years but those days are over. Without the snowpit data, the discussion of spatial variability is impossible, so section 3.4, Table 4 and all discussion about spatial variability should be removed from the paper.

2. The authors seem to be under the impression that if they don't observe negative nss sulfate, then there is no fractionation and no sea ice source. Of course this is not correct: while sea ice fractionation removes sulfate and causes negative nss-sulfate values, biogenic sulfate gives positive nss-sulfate. Only if the former overwhelms the latter will net negative values be seen. At sites very near the coast where marine biogenic inputs are large, this makes diagnosing fractionation tricky. As a rough estimate, one can note that typical values of MSA/nss-sulfate in biogenic input are 20% (Legrand and Pasteur 1998). From that we can estimate for example that biogenic sulfate at BI could easily have contributed all the sulfate seen, so that fractionation must have occurred. Uncertainty on the MSA concentration and the ration MSA/nss-sulfate makes this calculation very uncertain, but just illustrates that any of these sites could be experiencing large proportions of fractionated aerosol. The nss-sulfate discussion is valuable but needs to be done in a much more sophisticated way.

3. The authors use the correlations between concentration or flux and snow accumulation rate to try to diagnose the deposition mechanism. This could have some value if interpreted sensibly. However for S100 (1950-2000), it is obvious that the main feature is an immense rise in Na and Cl (factor 6) accompanied by a small drop (perhaps 20%) in accumulation rate. The relationship between these two trends will dominate any correlation but a 20% drop in snowfall cannot in itself cause more than a 20% increase in concentration even if dry deposition dominates completely. One simply cannot learn about dry and wet deposition for this site: something else is overwhelming the situation

by causing a huge increase in sea salt to the site.

4. The something else is causing huge sea salt concentration increases after 1950. It cannot be a change in the source to the ice shelf as a whole, since KC doesn't see it. I feel I am missing crucial information to allow me to interpret this. The obvious explanation would be that S100 has been getting closer to the ice shelf edge since 1950. But the paper gives no glaciological information that would allow us to interpret that. My assumption would be that the ice front at S100 occasionally calves icebergs, and that S100 is moving forwards at 10s to 100s of m/yr. The authors need to check and discuss what happened between 1950 and 2000. Did the S100 site simply get nearer the ice front?

I'm afraid all these points call for a major rethink about the purpose and conclusions of the paper. I will discuss a few more detailed points, but clearly any revision will be close to a new paper (it's borderline between major revision and reject) and will need reviewing again.

Abstract, page 2, line 5. As discussed above, the authors cannot conclude about dry deposition from the method they used. The very high concentrations do suggest a very high atmospheric concentration above the site by the year 2000, which would likely be both wet and dry deposited (such local material would have large particle sizes so would deposit fast). But this cannot exactly be described as dry deposition in the conventional aerosol dry deposition sense.

Page 10. The MSA-nitrate connection is overdone here. They surely end up on the same PC mainly because they don't show the sea salt pattern. We are not shown data that would allow us to judge this. However, for sure trying to link nitrate to MSA as a fertiliser seems far-fetched for a number of reasons. The Southern Ocean is not generally considered to be nitrate-limited; it seems unlikely that nitrate in the ocean is dominated by local atmospheric deposition. If you want to make this point you need to show data that would make a convincing case that high nitrate really is associated with
high MSA.

Page 11, line 8: they don't all show a 6 fold increase.

Section 3.3. You are doing something very difficult here. Please start with a discussion about the caveats: that it is very difficult to reliably divide the annual layers into 4 sections of equal time so the uncertainty on this is very large.

Section 3.4 should be removed as discussed above. At the end of the section, you dismiss the importance of elevation, but this cannot be excluded as a factor for the 3 ice rises reaching 200 m.

Section 3.5 – see discussion above.

Section 3.6 is very confused. The best way to treat this is to use the slope of lines such as that in Fig 5 to estimate the degree of fractionation, rather than trying different ratios (0.06, 0.04, 0.02). However, the line in Fig 5 should be a best fit through all the data (not just the negative), and should not go through zero (because when there is no sea salt there is still nss-sulfate from biogenic sources). Treated this way, I guess the slope will be about -0.04, implying 66% fractionation for the whole dataset (not just the post 1950s unless you see a significantly different slope for the two time periods)
* * *

---

## Referee Comment (RC2) · Anonymous Referee #2 · 29 Aug 2017

Comments to the paper: C.P. Vega et al., Spatial and temporal variability of sea-salts in ice cores and snow pits from Fimbul Ice Shelf, Antarctica.

General Comments

The paper is concerning the study of sea-salt components in snow, firn and ice samples collected in a coastal area of East Antarctica. There is a lack of knowledge on environmental and climatic data from firn/ice core stratigraphies in Coastal Regions of Antarctica; therefore, the paper is interesting for the Antarctic Glaciology Community. The paper is well written and sufficiently concise (see Comments to address toward more synthetic sections). However, in my opinion, the manuscript contains

some weak points that should be addressed before to be accepted for publication on The Cryosphere journal. These weak points are discussed in the Specific and Minor Comments section, but can be here listed:

1. some experimental procedures should be clarified; 2. ss- and nss- fractions of most of the analyzed components (especially Na and Ca) should be calculated as more reliable markers of sea spray (ssNa) and crustal (nssCa) contributions; 3. seasonal characterization of the sub-samples should be made taking into account the d18O profiles, instead of using an interpolation procedure; 4. the evaluation of the spatial variability by snow pit data appears to be not significant, because of the short record (lower than 1-year deposition), 5. the evaluation of ss-sulfate depletion from negative nss-sulfate values has to be completely revised; 6. I'm not convinced about the explanation of abrupt changes in sea salt deposition since 1950 in the S100 ice core (see, Specific Comments);

For these reasons, in my opinion, the manuscript needs major revisions before to be published on the The Cryosphere journal.

Specific and Minor Comments

Title: I'd suggest adding the Antarctic Region where Fimbul Ice Shelf is located (I think DML – Dronning Maud Land). Besides, since three shallow firn cores were analyzed, I'd suggest changing "ice cores" in "ice/firn cores".

Abstract. Authors should add some basic information about the sampling area.

Line 22, page 1. Please, change "three firn cores" in "three shallow firn cores (about 20 m deep)"

Lines 22-23, page 1. Please, add "(Dronning Maud Land – DML) to "Fimbul Ice Shelf (FIS)" location.

Line 24, page 1. Please, change "five snow pits" in "five snow pits (60-90 cm deep)"

Line 27, page 1. Please, change "elevation and distance to the sea" in "elevation (50-400 m a.s.l.) and distance (3-117 km) to the sea.

Lines 28-29, page 1. As the same Authors say at Lines 6-7, page 13, latitude and distance from the sea are related one to the other. I'd suggest referring just to the distance from the sea, as the most significant parameter (other than altitude) for the site characterization.

Lines 7-9 page 2. See Specific Comments for the interpretation of the S100 changes in sea salt deposition.

Line 9, page 2. Please change "ice rises cores" in "firn rises cores".

Introduction. This section seems to be too long and contains several information well known to the Glaciology Community. I'd suggest to summarize such information (especially those related to sea salt sources, specifically discussed in Section 4) focusing on the specific features of low altitude coastal sites, located in areas characterized by the presence of ice rises and ice rumples.

Lines 2-3, page 4. I'd suggest changing "This evidence . . .. . . values . . ..) in "This hypothesis is supported by the experimental evidence that the original seawater SO4/Na ratio cannot be used in nss-SO4 calculation, leading to negative values . . ..".

Lines 7-8, page 4. I'd suggest changing this sentence in: "These negative values indicate that a lower SO4/Na ratio has to be used in nss-SO4 calculation; i.e., a depletion of SO4, with respect to seawater composition, occurred in wet and dry depositions."

Lines 9, page 4 – Line 15, page 5. In my opinion, the discussion about the formation of brine, frost flower and other possible sources of fractionated sea-salt aerosol, although interesting, is too long in the introduction. This part should be summarized, eventually moving some key sentences in the Discussion Section.

Line 14, page 4. I think that frost flower, due their fragile structure, cannot cover the fractionated brine, but constitute an alternative processes leading the sulfate fractionation (as successively well explained by the Authors). I would suggest deleting "frost flower" in this sentence.

Line 6, page 6. Authors should give some basic information on the ranges of altitude and distance from the sea of the FIS ice rises (even if specific data are reported in Table 1). For instance: "Several ice rises (250-400 m a.s.l.; 10-50 km from the sea) are found at FIS …..".

Line 30, page 6. Please, add "about 20 m deep" after "Three shallow firn cores".

Lines 8-9, page 7. I think 4-8 cm is the sample resolution for firn cores. Authors should clarify that.

Lines 11-13, page 7. Information on the sample resolution for the S100 ice core should be reported.

Line 14, page 7. Please, change "five snow pits …" in "five snow pits (60-90 cm deep)…".

Line 20, page 7. I'd suggest adding: ", therefore snow pits samples cover the last year deposition".

Line 22, page 7. There is a reason why ammonium was not determined together with the other cations?

Line 4, page 8. Here, a resolution of 2 cm is indicated, while Lines 17-18 report a resolution of 4 cm. Authors are requested to clarify the resolution of G4a and G5a snow pits.

Lines 14-16, page 8. I'm aware that Na probably originates mainly from sea salt in this coastal area. However, Authors should use ss-Na as specific sea salt marker or justify the choice of using total Na by demonstrating that nss-Na is a negligible (for instance, lower than 5%) fraction of total Na. This could be especially relevant for KC site, where PCA shows a significant crustal contribution.

Line 4 and Line 8, page 9. How the Authors identified the previous summer layers, if 18dO measurements were not performed? By ice lens, different density, or other physical features? Authors should clarify their procedure, even if a reference is cited.

Section 3.1. Firn cores and snow pits values are reported as median. This is correct but it was not possible to evaluate the data variability, in order to compare the different sea salt contributes in the different sampling site. If median is used, then 25th and 75th percentile have to be shown, at least. I'd suggest plotting box plots with median, percentiles and outlier for each data set and reporting mean, minimum, maximum, and standard deviation in Table 2. In this way, it will be easier to appreciate the significance of the inter-site comparison.

Line 4, page 10 and following. PCA analysis. Usually, PCA analysis is carried out on raw data. Authors should clarify why They used normalized values as input for PCA data matrices. Indeed, PCA analysis on raw data is able to give results independent site-by-site and comparable among them. I do not know if this can be a result of the normalization, but the factor loading in every PCx factor seems to be quite low (lower than 0.5 in the majority of the loadings). In every way, I agree with the PCA results: the factors are surely related to sea salt, biogenic emission (mixed to nitrate) and crustal (for the site farthest from the sea) sources.

Lines 26-27, page 10. As a marker of the crustal source, the nss-Ca fraction has to be calculated at least for the KC site (but it could be useful to evaluate the ss- and nss-fractions for all the components in every data record).

Line 29-30, page 10. A further explanation could be common transport processes or pathways from marine areas at lower latitude. I do not think that nitrate, as a major nutrient, is a limiting factor for phytoplanktonic bloom in the Antarctic marine regions.

Line 8, page 11. The common sea-salt source between Cl and Na has to be confirmed (other than from PCA analysis) by calculating the Cl/ssNa ratios and comparing them with seawater composition. In this way, also a possible chloride depletion (for instance,

by wet or dry deposition of aged sea spray aerosol) could be observed. In particular, it has to be noted and discussed that Na and Cl have quite different temporal profiles in the KC firn core (constant or light increase for Cl; quite sharp decrease for Na).

Section 3.3. Seasonal pattern. I have some doubts about the linear interpolation procedure. Indeed, a simulated resolution of about 3 days has not a physical meaning, especially considering the variability of composition and temporal occurrence and frequency of snowfall events and dry deposition. Have the Authors information about the seasonal pattern of snowfalls in the studied area? I strongly suggest attributing the sample seasonality by using the raw data and the d18O profiles, identifying the four seasons by high, low or intermediate d18O values or, simply, classifying the samples as "summer" or "winter" samples, by assuming a threshold for summer/winter d18O values. If the results of this seasonal attribution are different from those reported in figure 3, all the section should be revised accordingly.

Section 3.4. Spatial variability. As the same Authors say (Line 6-7, page 13), differences in latitude reflect differences in distance from the coast. Besides, longitude variations are too little (at least for snow pits) to constitute a significant parameter for ion composition (as demonstrated by the not significant relationships, see Line 8-9, page 13). In the studied area, I think the only significant parameter is the distance from the sea and, possibly, the altitude (at least for the firn cores). Therefore, I suggest to discuss only the effect of the distance from the sea (and altitude and position with respect to the glacier tongue, if firn cores are included in the discussion) in this section.

Line 5, page 13. Probably, all these ions are mainly (or completely) coming from sea spray. It could be interesting to calculate their ss- and nss- fractions (at least for Na, Ca and Mg).

Section 3.6. Authors have to be aware that the evaluation of sulfate depletion from the observation of negative values of nss-sulfate is a difficult and controversial task. In coastal areas, the contribution of nss-SO4 from phytoplanktonic emissions (marked by

relatively high MSA concentrations) could be very relevant in spring/summer period. The nss-SO4 originated by the biogenic source can "cover" the ss-sulfate depletion by adding nss-SO4 to the sulfate budget. Therefore, a ss-sulfate depletion can occur even if no negative nss-SO4 values were found. The correction for biogenic nss-sulfate can be made by considering aerosol size distribution, aerosol seasonality and the nssSO4/MSA ratio from DMS atmospheric oxidation. No discussion on this relevant topic is given in this manuscript.

Line 23, page 13. Please, change: ". . . increase in ion concentrations after 1950s . . . " in ". . . increase in ion concentrations after 1950s in the S100 ice core. . .".

Line 30, page 13. Nitrate and sulfate correlation coefficient can be statistically significant, probably thanks to the high number of samples, but their values are so low (-0.04 and -0.10) to exclude every real correlation.

Lines 11-12, page 14. A dominant role of dry deposition should be demonstrated also by a significant negative correlation between snow concentration and accumulation rate. By looking to Table 5, that does not occur for Na (Rconc = -0.12). I think that both wet and dry deposition are relevant for a site so near to the sea (3 km) and located at so low altitude (48 m a.s.l.) to be affected from snowfall deposition, as well as from direct sea-spray primary production (i.e., aerosol directly produced by wind action on open sea surface or on frost flowers/brine structures over the sea ice surface).

Line 25, page 14. The value 0.06 is the well-known SO4/Na molar ratio in seawater (corresponding to the w/w ratio of 0.25) and the reference here cited is not pertinent.

Line 29, page 14. the negative values do not mean that "nssSO4 is strongly depleted in SO4 relative to Na", but that the sea-salt content in the snow is depleted in ss-sulfate with respect to seawater composition, so leading to an under-evaluation of nssSO4, if the 0.06 SO4/Na ratio is used. Indeed, if part of the original ss-sulfate is precipitated as mirabilite (in case of frost flower formation), the sea salt aerosol originated from the sea-ice surface is depleted in seawater sulfate.

Line 1, page 15 and following. In order to better understand the meaning of the lower SO4/Na (k) values here calculated, the Authors are requested to compare these values (k', k", k"') with the SO4/Na ratio for the frost flowers (0.07 w/w, Wagenbach et al., 1998, corresponding to 0.017 mol/mol). In this way, Authors could evaluate the relative contributions of sea salt from seawater (k = 0 0.06) and from frost flowers (k = 0.017).

Section 4. Discussion. The discussion is focused on the interpretation of the temporal trend of sea-salt components in the S100 ice core, showing a dramatic increase from 1950 to 2000. Indeed, this increase is impressive and, in my knowledge, never observed (with this intensity) in coastal and inner area of Antarctica (with the exception, of course, of the interglacial/glacial changes). Authors report a 6-times concentration increase from 1750-1949 to 1950-2000 periods. This difference, however, is calculate on the long-period median values (also in this case, percentile values or mean values with standard deviations could help in evaluating the data variability). If we observe the increasing trend of Na in figure 2, we can note that the concentration increases follows a continuous and quite constant trend, up to values as high as 200 umol/L. In comparison with the quite constant concentrations measured along the 1750-1949 period (around 15 umol/L), the Na concentration increases of a factor higher than 13. Chloride follows a similar trend. These S100 sea-salt values are about 4 to 20 times higher than those measured in the three firn cores. Authors attribute this large variation to changes in extension and persistence of sea ice east of the glacier tongue, after the tongue breaking on 1967. The decrease of the glacial tongue could have been against the preservation of multi-annual sea ice, promoting annual fast sea ice, where the formation of fractionated sea-salt aerosol and of frost flowers is more efficient (Lines 3-6, page 17). This is possible (and probable), but some experimental evidences, in my opinion, make weak this hypothesis. 1. The increase is not related to abrupt changes starting on 1967, but Figure 2 shows a very gradual and progressive trend (at least seeing the "linear trends" plotted in figures 2a and 2c) since 1950, before the glacier tongue breaking. 2. The two firn cores located on the same side of the glacier tongue (KM and KC) show sea-salt concentrations very lower and not characterized by an increasing trend (at least for the years covered by the records). In particular, the nearest fin core (KC) show stable (Cl) or slightly decreasing (Na) concentrations about 10 times lower than those measured in the most recent S100 sections (Figure 2b and 2d). 3. All the firn cores (and, in particular, KC and KM), do not show negative nss-sulfate values (Figure 4c). Authors attribute this different pattern to changes in altitude (lines 25-28, page 16), but KC, for instance, is only 200 m above the S100 site and particles as small as 1 um (Line 26, page 16) are easily transported to very high altitudes. If the altitude plays a similar dramatic effect, then no contribution of sea salt from frost flowers or salty snow sublimated aerosol should be observed in high-altitude plateau sites. On the contrary, some evidences of ss-sulfate depletion (shown by negative nss-SO4 values) have been found at the sites where Dome Fuji, EPICA-DML and EPICA-DC ice cores were drilled. In conclusion, in my opinion, the Authors hypothesis seems to be not confirmed by the firn cores data and other mechanisms should be investigated.

Lines 19-20, page 17. I strong suggest adding at least a composite figure showing the most relevant changes of extension and shape of the glacier tongue since 1967.

---

## Author Comment (AC1) · 30 Sep 2017

The authors (TA) would like to thank Referee 1 (R. 1) and Referee 2 (R. 2) for the time taken to review this manuscript. We value the referees' general and specific comments, and suggestions, and we will consider them in detail to prepare the revised version of this manuscript. Here we present our preliminary response to the main comments of R. 1 and R. 2.This response will be further expanded and detailed during the final response period.

Please also note the supplement to this comment:

[Figure]

https://www.the-cryosphere-discuss.net/tc-2017-148/tc-2017-148-AC1-supplement.pdf

[Figure]

**Supplement:**

**Response to Referee 1 comments**

The authors (TA) would like to thank Referee 1 (R. 1) and Referee 2 (R. 2) for the time taken to review this manuscript. We value the referees' general and specific comments, and suggestions, and we will consider them in detail to prepare the revised version of this manuscript. Here we present our preliminary response to the main comments of R. 1 and R. 2.This response will be further expanded and detailed during the final response period.

*R. 1: This paper presents new ice core chemical data for a coastal region of Antarctica. It interprets particularly the sea salt chemistry, and attempts to discuss the mechanisms behind sea salt production and deposition from the data. The data have some interest, particularly the unusual record from S100, and it may be possible to make a workable paper out of them. (The application of this paper is somewhat reduced because sites so close to the sea can be interesting but do not tell us too much about processes affecting inland sites.)*

**TA.:** The data presented in this manuscript contribute to fill in the gap in ice core records that exists for low elevation coastal areas. The importance of acquiring more data from the coastal areas is stressed by recent research (Stenni et al., 2017, and Thomas et al., 2017), and also the general comments of R. 2. We agree with R. 1 that those sites are interesting. However, information about inland sites cannot be expected from a study dealing with data from coastal sites only.

**Response to R. 1 points:**

*Point 1: The snow pit data are all from samples covering less than a year of snowfall. This makes it impossible to use the average values generated quantitatively, both because they are not a real yearly value, and because interannual variability means that the average for one year should have a huge uncertainty on it. There was a time when we were so desperate for new data from unexplored parts of the continent that we would at least consider surface snow data from part-years but those days are over. Without the snow pit data, the discussion of spatial variability is impossible, so section 3.4, Table 4 and all discussion about spatial variability should be removed from the paper.*

**TA.** We agree. We will discuss the spatial variability using only the ice rises and S100 cores, or as the referee suggested, eliminate the section.

*Point 2: The authors seem to be under the impression that if they don't observe negative nss sulfate, then there is no fractionation and no sea ice source. Of course this is not correct: while sea ice fractionation removes sulfate and causes negative nss-sulfate values, biogenic sulfate gives positive nss-sulfate. Only if the former overwhelms the latter will net negative values be seen. At sites very near the coast where marine biogenic inputs are large, this makes diagnosing fractionation tricky. As a rough estimate, one can note that typical values of MSA/mss-sulfate in biogenic input are 20% (Legrand and Pasteur 1998). From that we can estimate for example that biogenic sulfate at BI could easily have contributed all the sulfate seen, so that fractionation must have occurred. Uncertainty on the MSA concentration and the ration MSA/nss-sulfate makes this calculation very uncertain, but just illustrates that any of these sites could be experiencing large proportions of fractionated aerosol. The nss-sulfate discussion is valuable but needs to be done in a much more sophisticated way.*

**TA.** We do not think that no negative nss-sulfate values mean no fractionation. We will re-write the part in which we introduce the nss-sulfate findings so the message can be clearly delivered. We will also improve the discussion, including MSA and ss-fractions, as the referees suggested.

***Point 3:*** *The authors use the correlations between concentration or flux and snow accumulation rate to try to diagnose the deposition mechanism. This could have some value if interpreted sensibly. However for S100 (1950-2000), it is obvious that the main feature is an immense rise in Na and Cl (factor 6) accompanied by a small drop (perhaps 20%) in accumulation rate. The relationship between these two trends will dominate any correlation but a 20% drop in snowfall cannot in itself cause more than a 20% increase in concentration even if dry deposition dominates completely. One simply cannot learn about dry and wet deposition for this site: something else is overwhelming the situation by causing a huge increase in sea salt to the site.*

**TA.** We will check this part of the analysis. However, we do not understand the referee sentence "One simply cannot learn about dry and wet deposition for this site: something else is overwhelming the situation by causing a huge increase in sea salt to the site". Actually, because of this increase in $Na^+$ and $Cl^-$, we tried to constrain wet and dry deposition; if neither wet nor dry deposition is the cause of the increase in $Na^+$ and $Cl^-$ (as we understood from the referee comment), then what else can we causing the increase? We would be very grateful if the referee could clarify what she/he meant so we can improve the analysis following the referee's suggestion to this point.

***Point 4:*** *The something else is causing huge sea salt concentration increases after 1950. It cannot be a change in the source to the ice shelf as a whole, since KC doesn't see it. I feel I am missing crucial information to allow me to interpret this. The obvious explanation would be that S100 has been getting closer to the ice shelf edge since 1950. But the paper gives no glaciological information that would allow us to interpret that. My assumption would be that the ice front at S100 occasionally calves icebergs, and that S100 is moving forwards at 10s to 100s of m/yr. The authors need to check and discuss what happened between 1950 and 2000. Did the S100 site simply get nearer the ice front?*

**TA.** We didn't include more glaciological data because those have been published elsewhere. However, we agree with the referee's point and we will include relevant glaciological data of the area so the discussion can be clarified and better understood.

**Response to Referee 2 comments**

***Point 1:*** *some experimental procedures should be clarified*

**TA.** We will clarify the procedures as suggested by R. 2

***Point 2:*** *ss- and nss- fractions of most of the analyzed components (especially Na and Ca) should be calculated as more reliable markers of sea spray (ssNa) and crustal (nssCa) contributions.*

**TA.** Since we used $Na^+$ as reference ion for the calculation of nss-fractions, we assumed all $Na^+$ has a sea-salt origin, therefore, we cannot separate it into nss-$Na^+$ and ss-$Na^+$. We will present such apportioning for the other ions (Table A) in an additional table in the manuscript. In fact, we have done it in a previous version of the manuscript but we decided to let it out to focus on the

nssSO$_4^{2-}$ only and keep the paper concise. But we agree with the R. 2, and including the nss-fractions (and ss-fractions) for all ions will improve the discussion.

Table A. Median nss-ion concentrations (in μmol L$^{-1}$) in the KC, KM, BI, and S100 cores.

| Core | Period | nssCl$^-$ | nssSO$_4^{2-}$ | nssK$^+$ | nssMg$^{2+}$ | nssCa$^{2+}$ |
|------|--------|-----------|----------------|----------|--------------|--------------|
| | | | | *μmol L$^{-1}$* | | |
| KC | 1958–2007 | 0.1 | 1.1 | 1.8×10$^{-2}$ | 2.2×10$^{-2}$ | 0.3 |
| KM | 1995–2012 | 3.9 | 0.7 | 0.2 | 7.6×10$^{-3}$ | 0.4 |
| BI | 1996–2012 | 0.7 | 0.7 | 6.4×10$^{-2}$ | -3.8×10$^{-2}$ | 0.2 |
| S100 | 1737–2000 | -3.1 | -7.2×10$^{-2}$ | 3.4×10$^{-3}$ | -0.4 | 0.1 |
| S100 | 1995–2000 | -28.4 | -4.8 | -8.8×10$^{-4}$ | -5.5 | -5.4×10$^{-2}$ |
| S100 | 1737–1949 | -1.9 | 0.1 | -7.5×10$^{-4}$ | -0.2 | 0.2 |
| S100 | 1950–2000 | -18.3 | -3.1 | 4.4×10$^{-2}$ | -2.6 | 2.7×10$^{-2}$ |

***Point 3:*** *seasonal characterization of the sub-samples should be made taking into account the* $\delta^{18}O$ *profiles, instead of using an interpolation procedure.*

**TA.** We agree, we will re-work this section using the $\delta^{18}O$ maxima/minima chronology that we previously published. For that we will use the winter minima and summer maxima found in the $\delta^{18}O$ profiles for the KC and BI cores (Vega et al., 2016) (Figure A), and seasonalities in the $\delta^{18}O$ found in the S100 core (Kaczmarska et al., 2004), to obtain the subannual variability of the different ions, including ss-fractions and nss-fractions.

[Figure]

Figure A. Seasonality of (a, d) MSA, (b, e) Na$^+$, and (c, f) $\delta^{18}O$ for the KM and BI cores. Dashed lines and dotted lines indicate winter (summer). Figure from Vega et al. (2016).

***Point 4:*** *the evaluation of the spatial variability by snow pit data appears to be not significant, because of the short record (lower than 1-year deposition).*

**TA.** This was also pointed by R. 1, so we will consider removing this section or else, re-write it considering only the core data.

***Point 5:*** *the evaluation of ss-sulfate depletion from negative nss-sulfate values has to be completely revised.*

**TA.** We will do as suggested by R. 2 in the revised version of the manuscript.

***Point 6:*** *I'm not convinced about the explanation of abrupt changes in sea salt deposition since 1950 in the S100 ice core.*

**TA.** We will re-write the discussion. We will follow R. 1 and R. 2 comments and include glaciological data that will give more clarity to our explanation. Without satellite imagery before 1979, it is not possible to proof our hypothesis. It is a plausible explanation, however, there might be other things that have played an additional role.

**References**

Kaczmarska, M., Isaksson, E., Karlöf, L., Winther, J-G., Kohler, J., Godtliebsen, F., Ringstad Olsen, L., Hofstede, C. M., Van Den Broeke, M. R., Van De Wal, R. S.W., Gundestrup, N.: Accumulation variability derived from an ice core from coastal Dronning Maud Land, Antarctica, Ann. Glaciol. 39, 339–345, 2004.

Stenni, B., Curran, M. A. J., Abram, N. J., Orsi, A., Goursaud, S., Masson-Delmotte, V., Neukom, R., Goosse, H., Divine, D., van Ommen, T., Steig, E. J., Dixon, D. A., Thomas, E. R., Bertler, N. A. N., Isaksson, E., Ekaykin, A., Frezzotti, M., and Werner, M.: Antarctic climate variability at regional and continental scales over the last 2,000 years, Clim. Past Discuss., doi:10.5194/cp-2017-40, in review, 2017.

Thomas, E. R., van Wessem, J. M., Roberts, J., Isaksson, E., Schlosser, E., Fudge, T., Vallelonga, P., Medley, B., Lenaerts, J., Bertler, N., van den Broeke, M. R., Dixon, D. A., Frezzotti, M., Stenni, B., Curran, M., and Ekaykin, A. A.: Review of 23 regional Antarctic snow accumulation over the past 1000 years, Clim. Past Discuss., doi:10.5194/cp-2017-18, in review, 2017.

Vega, C. P., Schlosser, E., Divine, D. V., Kohler, J., Martma, T., Eichler, A., Schwikowski, M., and Isaksson, E.: Surface mass balance and water stable isotopes derived from firn cores on three ice rises, Fimbul Ice Shelf, Antarctica, The Cryosphere, 10, 2763–2777, doi:10.5194/tc-10-2763-2016, 2016.

---

## Author Response (AR1)

*Final author comments on "*Spatial and temporal variability of sea-salts in ice cores and snow pits from Fimbul Ice Shelf, Antarctica*" by C. P. Vega et al.*

To the referees:

The authors would like to thank the referees for the time they took to review this manuscript. The referees have provided valuable general and specific comments which have been very helpful to prepare a revised version of the manuscript. We agree in most of the referees comments, and consequently, we have done the correspondent modifications in the manuscript text to include their valuable suggestions, as long as it was possible. This letter contains a full response to the referees' comments, following a preliminary response uploaded on 2017-09-30 as part of the Discussion period.

We provide in this letter, a detailed point by point response to all referee comments. Referees comments are noted in italics, authors responses have been noted as *CV*. In addition, we provide a marked up version of the manuscript back-tracking the changes made, and a final version of the manuscript.

**Response to Anonymous Referee #1**

*This paper presents new ice core chemical data for a coastal region of Antarctica. It interprets particularly the sea salt chemistry, and attempts to discuss the mechanisms behind sea salt production and deposition from the data. The data have some interest, particularly the unusual record from S100, and it may be possible to make a workable paper out of them. (The application of this paper is somewhat reduced because sites so close to the sea can be interesting but do not tell us too much about processes affecting inland sites.)*

CV. As we have mentioned in the discussion period, the data presented in this manuscript contribute to fill in the existent gap in ice core records for low elevation coastal areas. The relevance of acquiring more data from the coastal areas has been stressed by recent research (Stenni et al., 2017, and Thomas et al., 2017), and it is also pointed out in the general comments by Referee #2. We agree with Referee #1 that those sites are interesting. However, information about inland sites cannot be expected from a study dealing with data from coastal sites only.

*However at the moment the paper suffers from three very major flaws, and an omission that render most of the interpretations dubious:*

CV. Points 1 to 3 have been addressed below. Referee #1 also mentions an 'omission' mostly associated to glaciological data between 1950s to 2000s. Relevant references for the glaciological settings of Fimbul Ice Shelf (FIS) have been included in the first version of the manuscript, however, we did not show the data in an explicit way when discussing the abrupt change in sea-salts after the 1950s. We have now included available glaciological data by Rignot et al. (2011) for FIS in the discussion part.

*1. The snowpit data are all from samples covering less than a year of snowfall. This makes it impossible to use the average values generated quantitatively, both because they are not a real yearly value, and because interannual variability means that the average for one year should have a huge uncertainty on it. There was a time when we were so desperate for new data from unexplored parts of the continent that we would at least consider surface snow data from part-years but those days are over. Without the snowpit data, the discussion of spatial variability is impossible, so section 3.4, Table 4 and all discussion about spatial variability should be removed from the paper.*

CV. The referee is correct about the snow pits use in the discussion in section 3.4; consequently, we have removed the snow pits from the manuscript main text and added them as Supplementary material. In order to keep the spatial variability discussion, we have now only used the core data (KC, KM, BI, and S100) for the years in which the cores overlap (1997–2000). We calculated R-values of the correlations between median annual concentrations for all ions versus latitude, longitude, site elevation, and distance from the sea. We found that only $SO_4^{2-}$ and MSA showed a significant (at the 95 % confidence interval) correlation with latitude, and longitude, respectively, whereas all other correlations in Table 4 were not significant. We attributed these results to the local effects on annual SMB evidenced at the KM and BI sites reported by Vega et al. (2016) which would affect any spatial pattern, e.g., the dependency reported by Stenberg et al. (1998) between sea-salts and distance from the sea.

*2. The authors seem to be under the impression that if they don't observe negative nss sulfate, then there is no fractionation and no sea ice source. Of course this is not correct: while sea ice fractionation removes sulfate and causes negative nss-sulfate values, biogenic sulfate gives positive nss-sulfate. Only if the former overwhelms the latter will net negative values be seen. At sites very near the coast where marine biogenic inputs are large, this makes diagnosing fractionation tricky. As a rough estimate, one can note that typical values of MSA/nss-sulfate in biogenic input are 20% (Legrand and Pasteur 1998). From that we can estimate for example that biogenic sulfate at BI could easily have contributed all the sulfate seen, so that fractionation must have occurred. Uncertainty on the MSA concentration and the ration MSA/nss-sulfate makes this calculation very uncertain, but just illustrates that any of these sites could be experiencing large proportions of fractionated aerosol. The nss-sulfate discussion is valuable but needs to be done in a much more sophisticated way.*

CV. As we mentioned in the preliminary response to the referees, we do not think that no negative nss-sulfate values mean no fractionation. In order to clarify this point and present a more solid discussion on the topic, we have calculated the $MSA/nssSO_4^{2-}$ ratio, and calculated the percentage of biogenic $nssSO_4^{2-}$ (bio-$nssSO_4^{2-}$) to total $SO_4^{2-}$ in all the cores (Table 7), and section 3.6 has been rewritten accordingly.

*3. The authors use the correlations between concentration or flux and snow accumulation rate to try to diagnose the deposition mechanism. This could have some value if interpreted sensibly. However for S100 (1950-2000), it is obvious that the main feature is an immense rise in Na and Cl (factor 6) accompanied by a small drop (perhaps 20%) in accumulation rate. The relationship between these two trends will dominate any correlation but a 20% drop in snowfall cannot in itself cause more than a 20% increase in concentration even if dry deposition dominates completely. One simply cannot learn about dry and wet deposition for this site: something else is overwhelming the situation by causing a huge increase in sea salt to the site.*

CV. The referee is correct. Consequently, we have removed the section and we focused more into explaining both the six-fold increase in sea-salts and the fractionated $nssSO_4^{2-}$ in terms of an enhancing of sea-salts originated from the blowing of salty snow over sea ice. We have then introduced the above in sections 3.6 and 4.

*4. The something else is causing huge sea salt concentration increases after 1950. It cannot be a change in the source to the ice shelf as a whole, since KC doesn't see it. I feel I am missing crucial information to allow me to interpret this. The obvious explanation would be that S100 has been getting closer to the ice shelf edge since 1950. But the paper gives no glaciological information that would allow us to interpret that. My assumption would be that the ice front at S100 occasionally calves icebergs, and that S100 is moving forwards at 10s to 100s of m/yr. The authors need to check and discuss what happened between 1950 and 2000. Did the S100 site simply get nearer the ice front?*

CV. We have now included ice speed information (m y$^{-1}$) available for FIS. Ice velocities near S100 are in the order of 10s–100s m y$^{-1}$ (Rignot et al., 2011) for the period 2007–2009. However, we cannot assume the same ice velocities for S100 for the pre-1950 and post-1950 periods. By using the S100 core sea-salt results we could hypothesize that the increase observed in sea-salts from 1950 could be linked to an increase in ice velocities in comparison to the 1737–1949 period; and that the calving event at Trolltunga in 1967 (Vinje, 1975), enhanced the input of fractionated sea-salts to the S100 core by modifying the sea-ice conditions around S100. This could be supported by the fact that negative $nssSO_4^{2-}$ values slowly decreased between 1950–1966, showing a marked minimum around 1966 (±3 years), which could have been induced by the calving event. We discuss this in section 4.

*I'm afraid all these points call for a major rethink about the purpose and conclusions of the paper. I will discuss a few more detailed points, but clearly any revision will be close to a new paper (it's borderline between major revision and reject) and will need reviewing again.*
*Abstract, page 2, line 5. As discussed above, the authors cannot conclude about dry deposition from the method they used. The very high concentrations do suggest a very high atmospheric concentration above the site by the year 2000, which would likely be both wet and dry deposited (such local material would have large particle sizes so would deposit fast). But this cannot exactly be described as dry deposition in the conventional aerosol dry deposition sense.*

CV. We appreciate the referee's suggestions. We consider that after the modifications done to the manuscript, this has now improved and the hypothesis of sea-ice sea-salts as source to the S100 is now described in clearer way.

*Page 10. The MSA-nitrate connection is overdone here. They surely end up on the same PC mainly because they don't show the sea salt pattern. We are not shown data that would allow us to judge this. However, for sure trying to link nitrate to MSA as a fertiliser seems far-fetched for a number of reasons. The Southern Ocean is not generally considered to be nitrate-limited; it seems unlikely that nitrate in the ocean is dominated by local atmospheric deposition. If you want to make this point you need to show data that would make a convincing case that high nitrate really is associated with high MSA.*

CV. We have stated in the paragraph that MSA and $NO_3^-$ have also shown coherence in other cores (Wendl et al., 2015), and that a fertilizer effect was proposed to explain such coherence in those Svalbard cores. We then briefly explained Wendl et al. (2015) hypothesis for the MSA and $NO_3^-$ coherence, and then we added "However, there is a variety of possible $NO_3^-$ sources and the relative importance of these sources at certain locations and time is still in discussion (Mulvaney and Wolff, 1993; Savarino et al., 2007; Wolff et al. 2008; Weller et al. 2011; Pasteris et al., 2014; Sofen et al. 2014)." We did not attribute the coherence observed between MSA and $NO_3^-$ in our cores (PC2) to the "fertilizing effect" proposed by Wendl et al. (2015). We have now corrected the paragraph so our point is clearly made. The sentence now reads: "High loadings of $NO_3^-$ and MSA in PC2, and thus, coherence between both species, have been observed in an ice core from Lomonosovfonna, Svalbard (Wendl et al., 2015), and a fertilizing effect was proposed as explanation for those findings. Wendl et al. (2015) suggest that enhanced atmospheric $NO_3^-$ concentrations and the corresponding nitrogen input to the ocean can trigger the growth of dimethyl-sulfide-(DMS)-producing phytoplankton. However, there is a variety of possible $NO_3^-$ sources to polar sites, and the relative importance of these sources at certain locations and time is still in discussion (Mulvaney and Wolff, 1993; Savarino et al., 2007; Wolff et al. 2008; Weller et al. 2011; Pasteris et al., 2014; Sofen et al. 2014)."

*Page 11, line 8: they don't all show a 6 fold increase.*

CV. The sentence has now been corrected and now reads: "In the S100 core, Na$^+$, Cl$^-$, K$^+$, and Mg$^{2+}$ median concentrations show a marked six-fold increase after the 1950s."

*Section 3.3. You are doing something very difficult here. Please start with a discussion about the caveats: that it is very difficult to reliably divide the annual layers into 4 sections of equal time so the uncertainty on this is very large.*

CV. This is a good point, and had also been mentioned by referee #2 (Point 3). Consequently, we have now followed both referees' suggestions and section 3.3 has been modified accordingly (please refer to response to referee #2 for more details on this section).

*Section 3.4 should be removed as discussed above. At the end of the section, you dismiss the importance of elevation, but this cannot be excluded as a factor for the 3 ice rises reaching 200 m.*

CV. We have now modified this section, as mentioned in the response to Point 1 of referee #1.

*Section 3.5 – see discussion above.*

*Section 3.6 is very confused. The best way to treat this is to use the slope of lines such as that in Fig 5 to estimate the degree of fractionation, rather than trying different ratios (0.06, 0.04, 0.02). However, the line in Fig 5 should be a best fit through all the data (not just the negative), and should not go through zero (because when there is no sea salt there is still nss-sulfate from biogenic sources). Treated this way, I guess the slope will be about -0.04, implying 66% fractionation for the whole dataset (not just the post 1950s unless you see a significantly different slope for the two time periods).*

CV. We have now followed the referee's suggestions and we have modified section 3.6 and the discussion, accordingly.

**Anonymous Referee #2**

**General Comments**
*The paper is concerning the study of sea-salt components in snow, firn and ice samples collected in a coastal area of East Antarctica. There is a lack of knowledge on environmental and climatic data from firn/ice core stratigraphies in Coastal Regions of Antarctica; therefore, the paper is interesting for the Antarctic Glaciology Community. The paper is well written and sufficiently concise (see Comments to address toward more synthetic sections). However, in my opinion, the manuscript contains some weak points that should be addressed before to be accepted for publication on The Cryosphere journal. These weak points are discussed in the Specific and Minor Comments section, but can be here listed:*

*1. Some experimental procedures should be clarified;*

CV. We have addressed this general comment on the responses to the referee's minor comments.

*2. Ss- and nss- fractions of most of the analyzed components (especially Na and Ca) should be calculated as more reliable markers of sea spray (ssNa) and crustal (nssCa) contributions;*

CV. We have now calculated $nssNa^+$ using $Ca^{2+}$ as reference, and assigning a crustal origin for the $nssNa^+$ in the cores. We then calculated the percentage of $nssNa^+$ to total $Na^+$, and estimated $ssNa^+$ removing the nss-fraction from the total $Na^+$. This is now presented in section 2.3 and discussed in section 3.5. The motivation to use $Ca^{2+}$ as a reference ion to obtain $nssNa^+$ instead of using the $Na^+/Cl^-$ ratio in seawater, was on the basis of the lower $Cl^-/Na^+$ ratios found in the S100 core in comparison to standard seawater (presented in Table 5, and discussed in section 3.5).
We then calculated both ss-and nss-fractions for $Cl^-$, $SO_4^{2-}$, $K^+$, and, $Mg^{2+}$ using $ssNa^+$ as reference ion and $k$ values obtained from standard mean seawater composition (Table S2 in the Supplementary material), and the calculation has been explained in section 2.3. Due to the low concentrations of $NO_3^-$ in standard seawater (Summerhayes and Thorpe, 1996), we did not separate $NO_3^-$ into nss- and ss-fractions (i.e., $NO_3^-$ is assumed to only have a nss-origin, as well as MSA). Median, mean, maximum, minimum, and standard deviation of nss- and ss-fractions are shown in Table 5 and Table 6.

*3. Seasonal characterization of the sub-samples should be made taking into account the $\delta^{18}O$ profiles, instead of using an interpolation procedure;*

CV. This point was also addressed by referee #1. We have now corrected this section following both referees' suggestions (please refer to the response to referee #2 in the section *Specific and Minor Comments* for more details on this procedure or directly to section 3.3 in the manuscript).

*4. The evaluation of the spatial variability by snow pit data appears to be not significant, because of the short record (lower than 1-year deposition),*

CV. Referee #1 also pinpointed this. In order to keep the spatial variability discussion, we have now only used the core data (KC, KM, BI, and S100) for the years in which the cores overlap (1997–2000) (please refer to response to point 1 of referee #1 for more details).

*5. The evaluation of ss-sulfate depletion from negative nss-sulfate values has to be completely revised;*

CV. This has been now done in section 3.6. We have also included a new section (3.5) in which we discuss the ss- and nss-fractions.

*6. I'm not convinced about the explanation of abrupt changes in sea salt deposition since 1950 in the S100 ice core (see, Specific Comments);*

CV. We have now improved our discussion around this point as suggested by both referees. For more details on this, please refer to our response to point 4 of referee #1, and to section 4 which has been rewritten accordingly.

*For these reasons, in my opinion, the manuscript needs major revisions before to be published on the The Cryosphere journal.*

CV. As we mentioned to referee #1, we consider that after the referees's comments and suggestions, the manuscript has now improved and the hypothesis of sea-ice sea-salts as source to the S100 is now described in clearer way.

**Specific and Minor Comments**
*Title: I'd suggest adding the Antarctic Region where Fimbul Ice Shelf is located (I think DML – Dronning Maud Land). Besides, since three shallow firn cores were analyzed, I'd suggest changing "ice cores" in "ice/firn cores".*

CV. The title has now been corrected to: "Spatial and temporal variability of sea-salts in ice/firn cores from Fimbul Ice Shelf, Dronning Maud Land – DML, Antarctica".

*Abstract. Authors should add some basic information about the sampling area.*

CV. We have now modified the first part of the abstract, and now it reads: "Major ions were analysed in firn/ice cores located at Fimbul Ice Shelf (FIS), Dronning Maud Land – DML, Antarctica. FIS is the largest ice shelf in the Haakon VII Sea, with an extent of approximately 36 500 km$^2$. Three shallow firn cores (about 20 m deep) were retrieved in different ice-rises, Kupol Ciolkovskogo (KC), Kupol Moskovskij (KM), and Blåskimen Island (BI), while a 100 m long core (S100) was drilled near the FIS edge."

*Line 22, page 1. Please, change "three firn cores" in "three shallow firn cores (about 20 m deep)*

CV. Since we modified the first part of the abstract, we have included this suggestion when we refer only to the ice-rises cores (please see response to the previous comment).

*Lines 22-23, page 1. Please, add "(Dronning Maud Land – DML) to "Fimbul Ice Shelf (FIS)" location.*

CV. This has been added.

*Line 24, page 1. Please, change "five snow pits" in "five snow pits (60-90 cm deep)"*

CV. Since we have removed the snow pit data from the main text (please refer to response to point 1 of referee #1 for more details), this has been modified and moved to the Supplementary material.

*Line 27, page 1. Please, change "elevation and distance to the sea" in "elevation (50- 400 m a.s.l.) and distance (3-117 km) to the sea.*

CV. Since we removed the snow pit data, the sentence now reads: "These sites are distributed over the entire FIS area so that they provide a variety of elevation (50–400 m a.s.l.) and distance (3–42 km) to the sea."

*Lines 28-29, page 1. As the same Authors say at Lines 6-7, page 13, latitude and distance from the sea are related one to the other. I'd suggest referring just to the distance from the sea, as the most significant parameter (other than altitude) for the site characterization.*

CV. We have now removed the sentence "Concentrations of these ions were found to decrease with latitude and distance from the sea." found in the abstract in view of the changes to section 3.4.

*Lines 7-9 page 2. See Specific Comments for the interpretation of the S100 changes in sea salt deposition.*

CV. This has been addressed in that part of the response letter. Section 4 in the manuscript has been changed accordingly.

*Line 9, page 2. Please change "ice rises cores" in "firn rises cores".*

CV. Here we referred to the cores drilled at the ice rises, therefore "ice rises cores" should be interpreted as the *cores drilled at the ice rises.* In order clarify the text, we have now changed "ice rises" for "ice-rises", when referring to the topographical feature.

Introduction. This section seems to be too long and contains several information well known to the Glaciology Community. I'd suggest to summarize such information (especially those related to sea salt sources, specifically discussed in Section 4) focusing on the specific features of low altitude coastal sites, located in areas characterized by the presence of ice rises and ice rumples.

CV. We consider that it is important to keep the information that it has been included in the introduction; therefore, we would like to keep it as it is.

*Lines 2-3, page 4. I'd suggest changing "This evidence : : :: : : values : : :.) in "This hypothesis is supported by the experimental evidence that the original seawater SO4/Na ratio cannot be used in nss-SO4 calculation, leading to negative values : : :.".*

CV. The sentence now reads: "This hypothesis is supported by the experimental evidence that the original seawater $SO_4^{2-}$/$Na^+$ ratio cannot be used in the non sea-salt sulfate ($nssSO_4^{2-}$) calculations, leading to negative $nssSO_4^{2-}$ values…"

*Lines 7-8, page 4. I'd suggest changing this sentence in: "These negative values indicate that a lower SO4/Na ratio has to be used in nss-SO4 calculation; i.e., a depletion of SO4, with respect to seawater composition, occurred in wet and dry depositions."*

CV. The sentence has now modified, and it now reads: "These negative values indicate that a lower $SO_4^{2-}$/$Na^+$ ratio has to be used in $nssSO_4^{2-}$ calculations, i.e., a depletion of $SO_4^{2-}$ with respect to seawater composition, occurred in wet and dry deposition."

*Lines 9, page 4 – Line 15, page 5. In my opinion, the discussion about the formation of brine, frost flower and other possible sources of fractionated sea-salt aerosol, although interesting, is too long in the introduction. This part should be summarized, eventually moving some key sentences in the Discussion Section.*

CV. We consider that it is important to keep this information as it is in the introduction to facilitate the understanding of the manuscript; therefore, we would like to keep it as it is.

*Line 14, page 4. I think that frost flower, due their fragile structure, cannot cover the fractionated brine, but constitute an alternative processes leading the sulfate fractionation (as successively well explained by the Authors). I would suggest deleting "frost flower" in this sentence.*

CV. "frost flower" has now been deleted from the sentence.

*Line 6, page 6. Authors should give some basic information on the ranges of altitude and distance from the sea of the FIS ice rises (even if specific data are reported in Table 1). For instance: "Several ice rises (250-400 m a.s.l.; 10-50 km from the sea) are found at FIS : : :...".*

CV. The sentence now reads: "Several ice-rises (250–400 m a.s.l.; 10–42 km from the coast) are found at FIS, varying in size from 15 to 1200 km$^2$, and located approximately 200 km apart."

*Line 30, page 6. Please, add "about 20 m deep" after "Three shallow firn cores".*

CV. This has been added.

*Lines 8-9, page 7. I think 4-8 cm is the sample resolution for firn cores. Authors should clarify that.*

CV. The referee is correct. We have changed "sample length" for "sample resolution".

*Lines 11-13, page 7. Information on the sample resolution for the S100 ice core should be reported.*

CV. We have now included the information as: "The S100 core was sampled at 5 cm resolution between top and 6 m deep, and then at 25 cm resolution between 6 m to 100 m deep."

*Line 14, page 7. Please, change "five snow pits : : :" in "five snow pits (60-90 cm deep): : :".*

CV. This has been now modified and moved to the Supplementary material.

*Line 20, page 7. I'd suggest adding: ", therefore snow pits samples cover the last year deposition".*

CV. This has now been added, and moved to the Supplementary material.

*Line 22, page 7. There is a reason why ammonium was not determined together with the other cations?*

CV. Ammonium was not measured in the S100 core due to analytical restrictions. Ammonium was measured in the KC, KM, and BI cores, and snow pits, but not reported because its concentrations were often close to the detection limit.

*Line 4, page 8. Here, a resolution of 2 cm is indicated, while Lines 17-18 report a resolution of 4 cm. Authors are requested to clarify the resolution of G4a and G5a snow pits.*

CV. The sample resolution of pits G4a y G5a was 4 cm. We have now corrected the sentence, and this part has been moved to the Supplementary material.

*Lines 14-16, page 8. I'm aware that Na probably originates mainly from sea salt in this coastal area. However, Authors should use ss-Na as specific sea salt marker or justify the choice of using total Na by demonstrating that nss-Na is a negligible (for instance, lower than 5%) fraction of total Na. This could be especially relevant for KC site, where PCA shows a significant crustal contribution.*

CV. The referee is correct. We have now included this in section 3.5 in which we discuss the $Cl^-/Na^+$ ratio, mechanisms of $Cl^-$ and $Na^+$ enrichment/depletion, and calculated $nssNa^+$, $ssNa^+$, and percentage of $nssNa^+$ to total $Na^+$, as suggested by the referee. For more details on this, please refer to the response given to point 2 of referee #2.

*Line 4 and Line 8, page 9. How the Authors identified the previous summer layers, if $\delta^{18}O$ measurements were not performed? By ice lens, different density, or other physical features? Authors should clarify their procedure, even if a reference is cited.*

CV. The estimation of the summer layer in each pit was visually and stratifically done by Sinisalo et al. (2013). We have now clarified the paragraph (and as metioned before, moved the snow pit data to the Supplementary material), and it now reads: "The previous summer layer in pits M1, M2 and G3–G5 was visually and stratifically identified at 120–160 cm depth according to Sinisalo et al. (2013); however, chemical sampling was done just down to a depth of 60–90 cm in each snow pit reported in this study, and samples for water stable isotope were not collected. Consequently, the snow depth at the different snow pit sites in which ions were analysed represents less than a year of accumulation (Table S1), and a highly precise dating (e.g. with monthly resolution) of the snow layers was not possible."

*Section 3.1. Firn cores and snow pits values are reported as median. This is correct but it was not possible to evaluate the data variability, in order to compare the different sea salt contributes in the different sampling site. If median is used, then 25th and 75th percentile have to be shown, at least. I'd suggest plotting box plots with median, percentiles and outlier for each data set and reporting mean, minimum, maximum, and standard deviation in Table 2. In this way, it will be easier to appreciate the significance of the inter-site comparison.*

CV. We have now included mean, minimum, maximum, and standard deviation in Table 2 for all ions in the different cores (and snow pits in Table S3). Box-plots for the different ions in the cores and snow pits have now been included as Figure S1–S3 in the Supplementary material.

*Line 4, page 10 and following. PCA analysis. Usually, PCA analysis is carried out on raw data. Authors should clarify why they used normalized values as input for PCA data matrices. Indeed, PCA analysis on raw data is able to give results independent site-by-site and comparable among them. I do not know if this can be a result of the normalization, but the factor loading in every PCx factor seems to be quite low (lower than 0.5 in the majority of the loadings). In every way, I agree with the PCA results: the factors are surely related to sea salt, biogenic emission (mixed to nitrate) and crustal (for the site farthest from the sea) sources.*

CV. The PCA technique calculates a new projection of the input data set on the basis of the standard deviation of each variable. Consequently, variables must be normalized so standard deviations are weighted equally when using PCA analysis.

*Lines 26-27, page 10. As a marker of the crustal source, the nss-Ca fraction has to be calculated at least for the KC site (but it could be useful to evaluate the ss- and nssfractions for all the components in every data record).*

CV. ss-fractions have been now calculated for $Cl^-$, $SO_4^{2-}$, $K^+$, $Mg^{2+}$, and $Na^+$ in all the cores, and presented in Table 5 and Table 6. The ss-fraction results are discussed in section 3.5. Please refer to our response to point 2 of referee #2 for more details.

*Line 29-30, page 10. A further explanation could be common transport processes or pathways from marine areas at lower latitude. I do not think that nitrate, as a major nutrient, is a limiting factor for phytoplanktonic bloom in the Antarctic marine regions.*

CV. Referee #1 has also commented on this. The paragraph has been re-written in view of the referees' comments (please refer to the response to referee #1 for more details).

*Line 8, page 11. The common sea-salt source between Cl and Na has to be confirmed (other than from PCA analysis) by calculating the Cl/ssNa ratios and comparing them with seawater composition. In this way, also a possible chloride depletion (for instance, by wet or dry deposition of aged sea spray aerosol) could be observed.*

CV. We have now included a discussion on this in a new section (3.5). For more details, please refer to our response to point 2 of referee #2.

*In particular, it has to be noted and discussed that Na and Cl have quite different temporal profiles in the KC firn core (constant or light increase for Cl; quite sharp decrease for Na).*

CV. Figure 2 b and d show the $Cl^-$ and $Na^+$ annual concentrations in the KC core. As noted in Table S4 the slopes of the linear regression for both ions are not significant at the 95 % confidence level, therefore we did not discuss the increase/decrease for $Cl^-$ and $Na^+$ in the KC core.

*Section 3.3. Seasonal pattern. I have some doubts about the linear interpolation procedure. Indeed, a simulated resolution of about 3 days has not a physical meaning, especially considering the variability of composition and temporal occurrence and frequency of snowfall events and dry deposition. Have the Authors information about the seasonal pattern of snowfalls in the studied area?*

CV. As we noted in the response to the referees of the paper by Vega et al. (2016) (https://www.the-cryosphere-discuss.net/tc-2016-164/tc-2016-164-AC1.pdf), extensive precipitation records at the core sites at FIS are, to our knowledge, not existent. The KC, and KM time scales used in this manuscript were constructed by Vega et al. (2016) based on $\delta^{18}O$ winter minima and summer maxima, and assuming uniform precipitation throughout the year at the core sites. This assumption was made on the basis of the precipitation regime at DML reported by Schlosser et al. (2008) which showed high temporal variability of the precipitation monthly sums due to the influence of cyclone activity affecting both coastal and inland regions. At Neumayer, the closest station to the core sites, two precipitation maxima are identifiable for the period 2001–2006 (April and October) possibly due to the semi-annual oscillation of the circumpolar trough (Schlosser el al., 2008). We have now included this information in at the beginning of section 3.3.

*I strongly suggest attributing the sample seasonality by using the raw data and the $\delta^{18}O$ profiles, identifying the four seasons by high, low or intermediate $\delta^{18}O$ values or, simply, classifying the samples as "summer" or "winter" samples, by assuming a threshold for summer/winter $\delta^{18}O$ values. If the results of this seasonal attribution are different from those reported in figure 3, all the section should be revised accordingly.*

CV. Both referees have pointed to this, therefore we have re-done the analysis considering $\delta^{18}O$ winter minima and summer maxima reported by Vega et al. (2016) as tie points for the seasonal analysis. We have then separated the annual samples into "winter" and "summer" samples, and we have rewritten section 3.3 accordingly.

*Section 3.4. Spatial variability. As the same Authors say (Line 6-7, page 13), differences in latitude reflect differences in distance from the coast. Besides, longitude variations are too little (at least for snow pits) to constitute a significant parameter for ion composition (as demonstrated by the not significant relationships, see Line 8-9, page 13). In the studied area, I think the only significant parameter is the distance from the sea and, possibly, the altitude (at least for the firn cores). Therefore, I suggest to discuss only the effect of the distance from the sea (and altitude and position with respect to the glacier tongue, if firn cores are included in the discussion) in this section.*

CV. Referee #1 also pinpointed this. In order to keep the spatial variability discussion, we have now only used the core data (KC, KM, BI, and S100) for the years in which the cores overlap (1997–2000). The discussion has been changed accordingly, with no significant relationship between the median annual ion concentrations and latitude, site elevation, and distance from the sea for any of the species (except for $SO_4^{2-}$ and MSA vs. latitude and longitude, respectively) (please refer to response to point 1 of referee #1 for more details).

*Line 5, page 13. Probably, all these ions are mainly (or completely) coming from sea spray. It could be interesting to calculate their ss- and nss- fractions (at least for Na, Ca and Mg).*

CV. ss-fractions have been now calculated for $Cl^-$, $SO_4^{2-}$, $K^+$, $Mg^{2+}$, and $Na^+$ in all the cores, and presented in Table 5 and Table 6. The ss-fraction results are discussed in section3.5.

*Section 3.6. Authors have to be aware that the evaluation of sulfate depletion from the observation of negative values of nss-sulfate is a difficult and controversial task. In coastal areas, the contribution of nss-SO4 from phytoplanktonic emissions (marked by relatively high MSA concentrations) could be very relevant in spring/summer period. The nss-SO4 originated by the biogenic source can "cover" the ss-sulfate depletion by adding nss-SO4 to the sulfate budget. Therefore, a ss-sulfate depletion can occur even if no negative nss-SO4 values were found. The correction for biogenic nsssulfate can be made by considering aerosol size distribution, aerosol seasonality and the nssSO4/MSA ratio from DMS atmospheric oxidation. No discussion on this relevant topic is given in this manuscript.*

CV. This was a common point between the referees. We followed the referees suggestions and in order to clarify this point and present a more solid discussion on the topic, we have calculated the

MSA/nssSO$_4^{2-}$ ratio, and calculated the percentage of biogenic nssSO$_4^{2-}$ (bio-nssSO$_4^{2-}$) to total SO$_4^{2-}$ in all the cores (Table 7), and section 3.6 has been rewritten accordingly.

*Line 23, page 13. Please, change: ": : : increase in ion concentrations after 1950s : : : " in ": : : increase in ion concentrations after 1950s in the S100 ice core: : :".*

CV. We removed the sentence in the revised version of the manuscript.

*Line 30, page 13. Nitrate and sulfate correlation coefficient can be statistically significant, probably thanks to the high number of samples, but their values are so low (-0.04 and -0.10) to exclude every real correlation.*

CV. We have removed the "deposition regime" section in the revised version of the manuscript after adding section 3.5 on sea-salt and non sea-salt fractions, and modified sections 3.6 and 4.

*Lines 11-12, page 14. A dominant role of dry deposition should be demonstrated also by a significant negative correlation between snow concentration and accumulation rate. By looking to Table 5, that does not occur for Na (Rconc = -0.12). I think that both wet and dry deposition are relevant for a site so near to the sea (3 km) and located at so low altitude (48 m a.s.l.) to be affected from snowfall deposition, as well as from direct sea-spray primary production (i.e., aerosol directly produced by wind action on open sea surface or on frost flowers/brine structures over the sea ice surface).*

CV.  As mentioned above, we removed the "deposition regime" section in the revised version of the manuscript.

*Line 25, page 14. The value 0.06 is the well-known SO4/Na molar ratio in seawater (corresponding to the w/w ratio of 0.25) and the reference here cited is not pertinent.*

CV. We have now referred to "Table S2, Supplementary material" for the current manuscript, and removed the Vega et al. (2015) reference.

*Line 29, page 14. the negative values do not mean that "nssSO4 is strongly depleted in SO4 relative to Na", but that the sea-salt content in the snow is depleted in ss-sulfate with respect to seawater composition, so leading to an under-evaluation of nssSO4, if the 0.06 SO4/Na ratio is used. Indeed, if part of the original sns-sulfate is precipitated as mirabilite (in case of frost flower formation), the sea salt aerosol originated from the sea-ice surface is depleted in seawater sulfate.*

CV. The referee is correct. We have now rewritten the sentence as: "These negative values found in the snow, i.e. the sea-salt content in snow is strongly depleted in ssSO$_4^{2-}$ relative to seawater composition,…".

*Line 1, page 15 and following. In order to better understand the meaning of the lower SO4/Na (k) values here calculated, the Authors are requested to compare these values (k', k", k"') with the SO4/Na ratio for the frost flowers (0.07 w/w, Wagenbach et al., 1998, corresponding to 0.017 mol/mol). In this way, Authors could evaluate the relative contributions of sea salt from seawater (k = 0 0.06) and from frost flowers (k = 0.017).*

CV. We have now included this discussion in section 3.6.

*Section 4. Discussion. The discussion is focused on the interpretation of the temporal trend of sea-salt components in the S100 ice core, showing a dramatic increase from 1950 to 2000. Indeed, this*

*increase is impressive and, in my knowledge, never observed (with this intensity) in coastal and inner area of Antarctica (with the exception, of course, of the interglacial/glacial changes). Authors report a 6-times concentration increase from 1750-1949 to 1950-2000 periods. This difference, however, is calculate on the long-period median values (also in this case, percentile values or mean values with standard deviations could help in evaluating the data variability).*

CV. We have now included mean, minimum, maximum, and standard deviation in Table 2 for all ions in the different cores (and snow pits in Table S3). Box-plots for the different ions in the cores and snow pits have now been included as Figure S1–S3 in the Supplementary material.

*If we observe the increasing trend of Na in figure 2, we can note that the concentration increases follows a continuous and quite constant trend, up to values as high as 200 umol/L. In comparison with the quite constant concentrations measured along the 1750-1949 period (around 15 umol/L), the Na concentration increases of a factor higher than 13. Chloride follows a similar trend. These S100 sea-salt values are about 4 to 20 times higher than those measured in the three firn cores. Authors attribute this large variation to changes in extension and persistence of sea ice east of the glacier tongue, after the tongue breaking on 1967. The decrease of the glacial tongue could have been against the preservation of multi-annual sea ice, promoting annual fast sea ice, where the formation of fractionated sea-salt aerosol and of frost flowers is more efficient (Lines 3-6, page 17). This is possible (and probable), but some experimental evidences, in my opinion, make weak this hypothesis. 1. The increase is not related to abrupt changes starting on 1967, but Figure 2 shows a very gradual and progressive trend (at least seeing the "linear trends" plotted in figures 2a and 2c) since 1950, before the glacier tongue breaking.*

CV. As suggested by the referees, we have now included some glaciological data that might help to support our hypothesis regarding the increase in sea-salt after 1950s, and the increase in the $ssSO_4^{2-}$ depletion factor in the 1950–2000 period in comparison to the 1737–1949 period. According to Rignot et al. (2011), ice velocities near S100 were in the order of 10s–100s m $y^{-1}$ for the period 2007–2009. However, we cannot assume the same ice velocities for S100 for the pre-1950 and post-1950 periods. By using the S100 core sea-salt results we could hypothesize that the increase observed in sea-salts from 1950 could be linked to an increase in ice velocities in comparison to the 1737–1949 period; and that the calving event at Trolltunga in 1967 (Vinje, 1975), enhanced the input of fractionated sea-salts to the S100 core by modifying the sea-ice conditions around S100. This could be supported by the fact that negative $nssSO_4^{2-}$ values slowly decreased between 1950–1966 showing a marked minimum around 1966 (±3 years), which could have been induced by the calving event.

*2. The two firn cores located on the same side of the glacier tongue (KM and KC) show sea-salt concentrations very lower and not characterized by an in-creasing trend (at least for the years covered by the records). In particular, the nearest fin core (KC) show stable (Cl) or slightly decreasing (Na) concentrations about 10 times lower than those measured in the most recent S100 sections (Figure 2b and 2d).*

CV. The referee is correct. However, it is difficult to explain, with the current data, the lower sea-salt values found at the ice-rises in comparison to the S100 site after the 1950s. We hypothesize that (section 4): "Considering that we found no correlation between ion concentrations and site elevation (section 3.4), a decrease in wind transport efficiency of frost flowers (size of 10–20 mm) and aerosol formed via (ii) (size >0.95 $\mu$m) (Seguin et al., 2014) due to increased elevation cannot be addressed to explain the lower sea-salt values observed at the ice-rises compared to the S100 site. As mentioned in section 3.4, local effects on annual SMB due to topography and meteorology at the KM and BI sites, reported by Vega et al. (2016), are most likely involved in the different load of sea-salt to these sites."

*3. All the firn cores (and, in particular, KC and KM), do not show negative nss-sulfate values (Figure 4c). Authors attribute this different pattern to changes in altitude (lines 25-28, page 16), but KC, for*

*instance, is only 200 m above the S100 site and particles as small as 1 um (Line 26, page 16) are easily transported to very high altitudes. If the altitude plays a similar dramatic effect, then no contribution of sea salt from frost flowers or salty snow sublimated aerosol should be observed in high-altitude plateau sites. On the contrary, some evidences of ss-sulfate depletion (shown by negative nss-SO4 values) have been found at the sites where Dome Fuji, EPICA-DML and EPICA-DC ice cores were drilled. In conclusion, in my opinion, the Authors hypothesis seems to be not confirmed by the firn cores data and other mechanisms should be investigated.*

CV. This point has now been corrected in view of the high percentage of bio-nssSO$_4^{2-}$ to total SO$_4^{2-}$ found for the ice-rises cores (Table 7). We have now included the following paragraph in section 3.6: "In the KM and BI cores, the estimation of bio-nssSO$_4^{2-}$ surpasses the total SO$_4^{2-}$ observed in these cores, while in the KC core the bio-nssSO$_4^{2-}$ would represent about 50% of total SO$_4^{2-}$. These high percentages were expected especially in the KC, and BI cores, in which the nssSO$_4^{2-}$ fraction dominates over ssSO$_4^{2-}$ (section 3.5). In the S100 core, bio-nssSO$_4^{2-}$ varies according to the time period considered with percentages three times higher during the period 1737–1749 (72 %), than the period 1950–2000 (24 %). It is important to bear in mind the estimation of bio-nssSO$_4^{2-}$ when assessing the possible effect of fractionated aerosols as a source of sea-salts to the snow. In the ice-rises cores, the high estimated bio-nssSO$_4^{2-}$ percentages would most likely mask any ssSO$_4^{2-}$ depletion in sea-salt aerosols, making fractionation hard to evidence; consequently, fewer negative nssSO$_4^{2-}$ values or the absence of them in the ice-rises cores would not directly indicate that there is no SO$_4^{2-}$ fractionation in sea-salt found in snow but rather reflect the dominance of bio-nssSO$_4^{2-}$ in these sites. In the S100 core, this could be relevant for the pre-1950 period in which estimated bio-nssSO$_4^{2-}$ accounts for 72 % of total SO$_4^{2-}$." We also added (in section 4): "The ice core data from S100 also suggest that there was a change in sea-salt deposition regime after the 1950s evidenced by an increase, up to six-fold, of median sea-salt concentrations after the 1950s in comparison with the previous 200 years. Although a negative trend in SMB has been observed in the S100 and KC cores for the second half of the 20$^{th}$ century (Figure 2e and f) (Vega et al., 2016), the 0.2 % m w.e. y$^{-1}$ decrease in accumulation registered in the S100 core after 1950 (Table S4) cannot account for the increase observed in sea-salt concentrations after 1950s. This increase in concentration is accompanied by a clear shift in nssSO$_4^{2-}$ toward negative values, indicative of ssSO$_4^{2-}$ depletion in sea-salts measured in the core in comparison to bulk seawater, with ssSO$_4^{2-}$ depletion factors of two for the period 1737–1949, and three for the period 1950–2000."

*Lines 19-20, page 17. I strong suggest adding at least a composite figure showing the most relevant changes of extension and shape of the glacier tongue since 1967.*

CV. Although the data needed to fulfil the referee´s request is available, i.e. front lines for the Trolltunga extent between 1952 and 2014, these data conforms a manuscript by J. van Oostveen (NPI) et al., which is currently in preparation, therefore, we cannot publish the dataset in the present manuscript. We consider that including the Corona Satellite data from 1963 and the present MODIS MOA image (from Quantarctica) is sufficient to show the abrupt change in Trolltunga, that took place within October 1963 and 1973 (J. van Oostveen, personal communication). For more information on the Trolltunga extent over the last decades, please contact J. van Oostveen (jelte.van.oostveen@npolar.no).

**References**

[revised manuscript text omitted]

Cont. Table 2

_Header (each cell lists five values):_ Median / Mean / Maximum / Minimum / σ (µmol L⁻¹)

[revised manuscript text omitted]

---

## Author Response (AR2)

*Letter to the Editor regarding the referees' comments on "*Spatial and temporal variability of sea-salts in ice cores and snow pits from Fimbul Ice Shelf, Antarctica*" by C. P. Vega et al.*

Dear Editor:

The authors would like to thank again the referees and the editor for the time taken to revise this manuscript. We agree with the referees comments and suggestions, and consequently, we have done the correspondent modifications in the manuscript text to include their valuable suggestions. We have now provided evidence that the ice shelf in front S100 has moved during the 1950–2000 period bringing the S100 site closer to the ice shelf edge. This was a major concern of referee #1 that we did not address in a previous revision of the manuscript because we did not clearly understand what exactly the petition of the referee was on this point. We have now considered this change in the S100 distance to the FIS edge in the paper discussion and conclusion (sections 4 and 5), and changed the abstract accordingly. We have also considered the correction of the ss- and nss-fractions as suggested by referee #2. In fact, using the equations system to obtain $ssNa^+$, $nssNa^+$, $ssCa^{2+}$, $nssCa^{2+}$ was simple to implement and all ss- and nss-fraction for major ions were re-calculated and corrected in the manuscript Tables, Figures and text. We appreciate the referee's suggestion regarding the calculation and we think that it has considerably improved the method section of the manuscript. In effect, as referee #2 mentioned, our previous assumption of $Ca^{2+}$ having only a crustal origin led us to overestimate $nssNa^+$, however, not in a significant way, and all recalculated median, mean, maxima, and minima ss- and nss- concentrations remained similar. Therefore, the conclusions of the manuscript remain the same after the recalculation of the fractions.

We have included in this letter, a detailed response to the referees' comments done in January 2018. Authors' responses are noted as **CV**. In addition, we provide a marked up version of the manuscript back-tracking the changes made, and a final version of the manuscript without tracked changes.

We consider that all the points made by referees have been covered in this revision, and we hope that you will consider this manuscript now as ready to be published in The Cryosphere.

Sincerely,

C. P. Vega, on behalf of all co-authors

**Response to Anonymous Referee #1**

**Ref. #1.** The authors have taken on board most of the comments I made on the earlier version, and although I don't always agree with the conclusions they reach (for example about nitrate as a fertiliser), most of them are now reasonable.

**CV.** We would like to point out that we have never mentioned nitrate as a fertilizer as a conclusion of the work presented in this manuscript (in any of its versions). We only have referred to previous work done in the Arctic by Wendl et al. (2015) in which they suggest a fertilizing effect of nitrate; those are not conclusions of this manuscript (Vega et al., in revision). We decided to mention Wendl et al. (2015) findings together with findings of other authors (i.e. Mulvaney and Wolff, 1993; Savarino et al., 2007; Wolff et al. 2008; Weller et al. 2011; Pasteris et al., 2014; Sofen et al. 2014) to stress the fact that the interpretation of nitrate concentrations in ice cores is a topic which is still in discussion. We have now re-written the paragraph in order to clearly state what it is explained above. The paragraph in section 3.1 now reads: "Table 3 shows high loadings of $NO_3^-$ and MSA in PC2, and thus, coherence between both species. This correspondence has been previously observed in an ice core from Lomonosovfonna, Svalbard, and a fertilizing effect was proposed as explanation for those findings (Wendl et al., 2015). Wendl et al. (2015) suggest that enhanced atmospheric $NO_3^-$ concentrations

and the corresponding nitrogen input to the ocean can trigger the growth of dimethyl-sulfide-(DMS)-producing phytoplankton. However, there is a variety of possible $NO_3^-$ sources to polar sites, and the relative importance of these sources at certain locations and time is still in discussion (Mulvaney and Wolff, 1993; Savarino et al., 2007; Wolff et al. 2008; Weller et al. 2011; Pasteris et al., 2014; Sofen et al. 2014)."

We hope that it is understood that we are not saying that nitrate has a fertilizing effect in the context of our results, but only refer to Wendl et al. (2015) in the context of nitrate source identification.

We therefore think that we are not reaching any conclusions that referee #1 would not agree with.

**Ref. #1.** The conclusion about the cause of the increasing concentration at S100 needs further work. Apart from that, the remaining issues with the paper are mainly to do with readability as the changes made are quite rambling and confusing and I suspect many readers will not be able to follow the arguments. In particular:

Section 3.4 and Table 4 should be deleted. The authors have agreed that the pit data cannot be used because they do not represent even a full year. This means that now they are trying to regress 4 data points against 4 variables which obviously cannot be done in any meaningful way. Even where you find significance it looks spurious: for example sulfate is significantly related to latitude because S100 and KM are higher values than the other two sites, but the latitude difference between them is a matter of minutes and it is not credible that this is actually what is causing the difference in sulfate. Simply state that you cannot see any significant relationship between chemistry and the variables assessed, although sea salt is clearly lowest at KC, which is further inland than the other sites.

**CV.** We agree with the points made by referee #1. Consequently, we have removed section 3.4 and Table 4 (appearing in the previous version of the manuscript). Since the section on ions spatial variability has now been removed, we consider that the title of the paper should be modified. We suggest the title *Variability of sea-salts in ice/firn cores from Fimbul Ice Shelf, Dronning Maud Land – DML, Antarctica*, and we are open to any suggestions the editor may have about it. Regarding ion spatial variability, we have kept the following lines as suggested by referee #1, now in section 3.1: "We found no significant relationship between the median annual ion concentration and latitude, site elevation, and distance from the sea for most of the species, with the exception of annual $SO_4^{2-}$ and annual MSA concentrations which show a significant decrease (at the 95 % confidence level) with latitude, and east longitude, respectively. However, additional ice cores from Fimbul are needed to obtain a robust conclusion on spatial distribution of major ions." We have also added the following sentence in parenthesis in section 3.1, as suggested by the referee: "In general, median concentrations in the KM core are higher than in the other ice-rises cores, e.g. six to eight-fold higher concentrations of $Na^+$, $K^+$, $Mg^{2+}$ and $Cl^-$ in the KM core than in the KC core (which is further inland than the other sites) are found for the period 1995–2012.".

**Ref. #1.** You have now added a lot of SI about the snowpits, which are not used in the text. I appreciate you probably put work into them and are attached to them, but they add nothing to the paper and complicate it for the reader. I propose removing Table S1, S3, and Figs S2 and S3, plus any mention of them in the text.

**CV.** We thought it would be adequate to keep the snow pit data in the supplementary material, but we agree with the referee that including the data might be confusing for the reader. Consequently, we have now removed the snow pit data from the supplementary material, and do not mention them in the main text anymore.

**Ref. #1.** The new tables 2, 5, 6 and 7 are unreadably complicated because of the way that different statistics are piled vertically on top of each other. I propose that the authors present these tables with only the mean values (or the medians if they prefer) in the main text. If they really want to show off the other statistics (which are really thesis material and not needed in the paper), then place them in the supplement.

**CV.** We have followed the referee's suggestion and we placed mean, maximum, minimum, and standard deviations found in tables 2, 5, 6, and 7, in the supplementary material. We only keep median values in Table 2, 4, 5, and 6, as the referee suggested.

**Ref. #1.** Sections 3.5 and 3.6 are very long and confusing. They need carefully evaluating by all authors who should then try and express the concepts more concisely, pointing carefully to the reduced Tables 5 and 7.

**CV.** Sections 3.5 and 3.6 are now sections 3.4 and 3.5, respectively. We have edited these sections as much as possible following the referees suggestions.

**Ref. #1.** The conclusion about the increase at S100 after 1950 is now more credible, but still ignores what seems the obvious answer, and the authors still did not address what I asked about. While it could be that the loss of the ice tongue somehow influenced sea ice formation and caused the change it is not obvious to me why this would affect S100 so strongly but not KC or KM. What I asked is whether the authors are sure that the ice shelf immediately in front of S100 has not retreated over the last 50 years, putting S100 much closer to the sea (something that cannot affect the fixed points at KM and KC). Although the authors point me to glaciological data there is nothing I can see that answers this question, because there is no information about the ice front position, but I can make a guess. S100 is apparently 3 km from the sea today. I would guess the ice front position is almost fixed relative to the ice rise adjacent to it, with regular calving events keeping up with the ice velocity behind it (high resolution images seem to show crevassing near the ice front so this seems likely). The authors say that the ice velocity is "10s-100s of m/a", but if one blows up the Rignot map the velocity in the free-running parts of Fimbul all appear to be in blue shades, ie at least 100 m/a. Then over 50 years the site S100 has moved at least 5km, and is now that much closer to the ice front, and out of any "shadow" from the unnamed ice rise next to it. A move from eg 8 km to 3 km from the sea might well be enough to explain the data. (As an example a paper by Gorlach et al (FRISP report 2, page 48) shows sea salt concentration increasing by a factor 4 over about 10 km near Neumayer, moving from what I estimate to be about 13 to 3 km from the ice edge).

**CV.** The referee points to a very important issue, and we have now included a contour line in Figure 1 that shows the extent of FIS in 1963, not only around Trolltunga (as in the previous version of the manuscript), but also down to the S100 site. As it can be seen in the FIS edge contours shown in Figure 1, by 1963 the FIS ice front in the vicinity of the S100 site was located about 17 km further north than at present. As the referee pointed out, S100 would also have got closer to the ice edge, moving at least 5 km closer to the sea over the 1950–2000 period. We argue that the calving of Trolltunga in 1967, the change in the ice front position and the ice advance in the vicinity of the S100 site led to the increase in sea-salts observed in the S100 core after 1950, but did not significantly change sea-salt concentrations in the ice-rises cores, which are located at fixed points on FIS. We have now included this hypothesis in the discussion and conclusions, and changed the abstract accordingly, and we think that the point made by the referee has now been addressed properly. We also would like to note that we have no access to the reference suggested by the referee, i.e. Gorlach et al (FRISP report 2, page 48) and therefore, we could not include it.

**Ref. #1.** This suggests that this section of the paper needs a further iteration as this seems a much simpler explanation than yours. I know that you are trying to explain a change in the nss-sulfate contribution as well, but this is actually poorly constrained. The nss sulfate is very obviously negative in the last 50 years because the sea salt is so high in concentration and dominates. In the pre-1950 period, the partitioning of sulfate between ss and nss is much less safe, and it remains plausible that the sea salt part may still be depleted. In any case whatever you choose to say, you cannot just ignore the likelihood that S100 is just a much more coastal site as time progresses.

**CV.** As we mentioned above, we think that we have now addressed the point made by the referee regarding the change in distance between the S100 site and the ice edge. Changes have been done accordingly in section 4, 5 and in the abstract.

**Response to Anonymous Referee # 2**

**Ref. #2.** I'm happy to see that Authors made almost all the changes I suggested and the text was improved accordingly. For this reason, I think that the manuscript is now ready to be published on The Cryosphere journal. However, I have to note a potentially relevant error in calculating the ss-and nss-fractions of Na and Ca. If I have well understood, Authors used totCa as a marker of the crustal source and calculated nss-Na by the Ca/Na ratio in the Earth crust. This is not correct because also Ca can have two main sources: mineral dust and sea spray. This is particularly true for coastal sites, where the ssCa contribution could be relevant or even dominant. In order to calculate more reliable ss- and nss-fractions of Na and Ca, Authors should use the following 4-equation system:

totNa = ssNa + nssNa
totCa = ssCa + nssCa
ssNa = totNa - 0.562 nssCa
nssCa = totCa - 0.038 ssNa

where 0.562 = Na+/Ca2+ (w/w) in the crust (Bowen, 1979), and 0.038 = Ca2+/Na+ (w/w) in seawater (Nozaki, 1997).
Bowen, H.J.M., 1979. Environmental Chemistry of the Elements. Academic Press, London.
Nozaki, Y., 1997. A fresh look at element distribution in the North Pacific. http:/www.agu.org/eos_elec/97025e.html.
By solving the equation system (4 equations, 4 unknown variables), a more correct evaluation of ss-Na can be obtained.
If Authors consider that all the Ca comes from the crustal sources, nssNa fraction can be over-estimated. As a consequence, the critical parameter ssNa can be significantly under-estimated and the error can propagate to the nss-SO4 calculation.
I'm aware that using the correct procedure to calculate a reliable ss-Na fraction could imply many changes in the manuscript figures and text, if ss-Na values obtained with the equation system are significantly different from those obtained by using totCa as crustal marker. Therefore, I'd suggest that the Authors test the differences in ss-Na calculation with the two different procedures, at least for a sub-dataset of samples in which the contribution of ss-Ca appears to be relevant (i.e., for samples enriched in sea salt, for which totCa cannot be used a-priori as crustal marker). If this test is positive (i.e., if the ss-Na values calculated with the two methods are quite close each other), Authors could leave unchanged figures and text. On the contrary, ss- and nss-fraction of all the components (Na, Ca, sulfate etc.) should be revised.

**CV.** We appreciate the correction made by the referee regarding the calculation of ssNa$^+$ and nssCa$^{2+}$. We have now implemented the equations system and therefore we recalculated all ss- and nss-fractions using this approach. We explain the methodology in section 2.3. We used seawater and crustal composition given by Summerhayes and Thorpe (1996), and Lutgens and Tarbuck (2012). As mentioned before in this letter, the fractions did not significantly change, and therefore the main results of the manuscript remained the same. We appreciate the referee's suggestion regarding the ss- and nss-fraction calculations, and we think that the methods section of the manuscript has been improved accordingly.

**References**

[revised manuscript text omitted]